# Mechanisms and function of de novo DNA methylation in placental development reveals an essential role for DNMT3B

Simon Andrews [1], Christel Krueger [1,2,10], Maravillas Mellado-Lopez[3], Myriam Hemberger [4,5,6], Wendy Dean [6,7], Vicente Perez-Garcia[3] & Courtney W. Hanna [2,8,9] ✉

DNA methylation is a repressive epigenetic modification that is essential for development, exemplified by the embryonic and perinatal lethality observed in mice lacking de novo DNA methyltransferases (DNMTs). Here we characterise the role for DNMT3A, 3B and 3L in gene regulation and development of the mouse placenta. We find that each DNMT establishes unique aspects of the placental methylome through targeting to distinct chromatin features. Loss of *Dnmt3b* results in de-repression of germline genes in trophoblast lineages and impaired formation of the maternal-foetal interface in the placental labyrinth. Using *Sox2*-Cre to delete *Dnmt3b* in the embryo, leaving expression intact in placental cells, the placental phenotype was rescued and, consequently, the embryonic lethality, as *Dnmt3b* null embryos could now survive to birth. We conclude that de novo DNA methylation by DNMT3B during embryogenesis is principally required to regulate placental development and function, which in turn is critical for embryo survival.

DNA methylation typically functions as a repressive epigenetic modification and has been extensively studied for its roles in development, health and disease. Functional investigations into the role of DNA methylation have taken place since the 1990's, yet establishing its role comprehensively in the mammalian placenta has not been undertaken. DNA methylation exerts its repressive function by sterically interfering with access for transcription factor-DNA interactions or by recruiting DNA methylation-binding repressor complexes to chromatin[1,2]. The gene silencing function of DNA methylation has been demonstrated for key regulatory regions conferring genomic imprinting and X chromosome inactivation, and at specific genomic targets such as CpG-rich repetitive elements and germline-specific genes[3]. Despite these specific regulatory roles, the majority of mammalian cells have high levels of DNA methylation across the entire genome, with the exception of CpG islands (CGIs) and active regulatory elements. There are two phases of development when widespread de novo DNA methylation occurs, during gametogenesis and post-implantation embryonic development. In both contexts, DNA methylation is conferred by de novo DNA methyltransferase (DNMT) 3A and 3B enzymes[4], with contributions in gametogenesis from catalytically inactive cofactor DNMT3L and rodent-specific DNMT3C[5,6]. Early knockout (KO) studies in mouse models have demonstrated that de novo DNA methylation is critical for development as *Dnmt3a*, *Dnmt3b* and *Dnmt3a/b* DKO embryos die in mid-gestation or shortly after birth[7]. However, recent mapping of embryonic lineage specification in de novo *Dnmt* KO embryos in exquisite detail using single-cell technologies did not identify dramatic effects on lineage commitment or cell identity in gastrulation-staged embryos[8]. Furthermore,

[1]Bioinformatics Programme, Babraham Institute, Cambridge, UK. [2]Epigenetics Programme, Babraham Institute, Cambridge, UK. [3]Centro de Investigación Príncipe Felipe, Valencia, Spain. [4]Department of Biochemistry and Molecular Biology, Cumming School of Medicine, University of Calgary, Calgary, AB, Canada. [5]Department of Medical Genetics, Cumming School of Medicine, University of Calgary, Calgary, AB, Canada. [6]Alberta Children's Hospital Research Institute, University of Calgary, Calgary, AB, Canada. [7]Department of Cell Biology and Anatomy, University of Calgary, Calgary, AB, Canada. [8]Department of Physiology, Development and Neuroscience, University of Cambridge, Cambridge, UK. [9]Centre for Trophoblast Research, University of Cambridge, Cambridge, UK. [10]Present address: Bioinformatics Innovation Hub, Altos Labs Cambridge Institute, Cambridge, UK. ✉e-mail: cwh36@cam.ac.uk

transcriptional de-repression in de novo *Dnmt* KO post-gastrulation embryos was largely restricted to germline genes and specific sub-classes of repetitive elements[9]. Thus, the molecular consequences underpinning the role for de novo DNA methylation in embryo survival remains an outstanding question.

Placental trophoblast cells have a unique genome-wide DNA methylation pattern, with the majority of the genome being spanned by large partially methylated domains (PMDs), unlike the highly methylated genomes of somatic cells. This enigmatic feature appears to be conserved among placental mammals and is maintained throughout pregnancy[10–12], and remarkably, is shared with many cancers[13]. While the importance of imprinted DNA methylation in placentation is well-established[14,15], the role for the genome-wide DNA methylation landscape acquired during placentation has yet to be established. In this study, we investigate the impact of de novo DNMTs on the trophoblast lineage during embryogenesis, revealing the importance of DNA methylation in genome regulation and placental development beyond genomic imprinting. Our findings show that the deposition of DNA methylation by DNMTs is influenced by the chromatin landscape in trophoblast cells and is not only essential for the formation of the maternal-foetal interface during placentation, but is critical for embryo survival.

## Results

### DNMT3A, 3B and 3L establish the trophoblast methylome

DNMT3A, 3B and 3L have been reported to contribute to DNA methylation at CGIs (defined as CpG-dense regions that comprise about 1% of the genome) in trophoblast cells[16]. To date, the role of de novo DNMTs in establishing the genome-wide DNA methylation landscape in trophoblast remains unknown. Studies in the mouse epiblast and visceral endoderm have shown that de novo DNA methylation occurs between embryonic days (E)4.5 and 7.5;[17] hence, we evaluated *Dnmt* expression and DNA methylation in E7.5 extra-embryonic ectoderm (ExE) comprising trophoblast cells, and contrasted these to E7.5 epiblast. Assessing the DNA methylation land-scape genome-wide is necessary to understand which genomic elements are targeted by specific DNMTs and how this DNA methylation may regulate the underlying genomic features.

Using published RNA-seq data[18], we found that *Dnmt3a, Dnmt3b* and *Dnmt3l* are all highly expressed in ExE, with *Dnmt3a* and *Dnmt3b* showing approximately two-fold lower expression than in the epiblast and *Dnmt3l* showing 20-fold higher expression (Supplementary Fig. 1a). Using low-input post bisulphite adaptor tagging (PBAT), we assayed DNA methylation in E7.5 ExE and epiblast from *Dnmt3a* KO, *Dnmt3b* KO, *Dnmt3a/b* double KO (DKO), *Dnmt3l* KO and wildtype (WT) controls (Fig. 1a–c, Supplementary Data 1). When comparing genome-wide profiles of DNA methylation in E7.5 ExE using 100-CpG windows, all KOs cluster distinctly from WT ExE (Fig. 1b, c). Differentially methylated regions (DMRs) were identified using logistic regression and >20% difference in DNA methylation, demonstrating that there was a loss of DNA methylation in the ExE of *Dnmt3a* KO, *Dnmt3l* KO and *Dnmt3b* KO at 0.7%, 34.6% and 58.9% of autosomal 100-CpG windows, respectively (Supplementary Fig. 1b). Notably, these differences were markedly more pronounced than those observed in the corresponding KO epiblast tissues (0.2%, 0.1%, and 25.3% DMRs, respectively) (Fig. 1b, c, Supplementary Fig. 1c). Paradoxically, DNMT3A appears to be simultaneously dispensable and yet remark-ably competent to de novo methylate; in other words, there was almost no loss of DNA methylation in the absence of DNMT3A, but, in the absence of DNMT3B, DNMT3A can still establish a significant amount of genome-wide DNA methylation (Fig. 1b, c). DNMT3L shows an ExE-specific role in facilitating de novo DNA methylation, as *Dnmt3l* KO epiblast clustered with WT while *Dnmt3l* KO ExE clustered near *Dnmt3b* KO by PCA (Fig. 1b). In *Dnmt3a/b* DKOs, both ExE and epiblast fail to establish tissue-specific DNA methylation patterns and hence

cluster together with publicly available data from WT E3.5 inner cell mass and trophectoderm samples[16] (Fig. 1b, c). These data support that all DNMTs are required for the trophoblast methylome, albeit at highly discrepant levels.

### Targeting of DNMTs is linked to the chromatin landscape

The de novo DNMT enzymes have been shown to interact with mod-ified histone tails, which modulates their catalytic activity and genome localisation in vitro[19–21], although evidence that histone modifications may influence DNMT targeting in vivo remains limited. Thus, we gen-erated ultra-low input ChIP-seq data for H3K4me1 and H3K27ac in E6.5 ExE to combine with published datasets for H3K4me3, H3K27me3 and H3K36me3[18], immediately preceding the completion of de novo DNA methylation (Fig. 2a, Supplementary Data 1). Using UMAP dimension-ality reduction and clustering, we identified chromatin features genome-wide for 100-CpG windows, which were then annotated based on abundance of histone marks within each cluster (Fig. 2b, c). Func-tional genomic features have well-defined epigenetics signatures[22] and, consistently, our annotated chromatin features robustly represent the expected underlying genomic state. H3K36me3-marked gene bodies were the most highly DNA methylated, whereas active promoters were predominantly unmethylated in E7.5 WT ExE (Supplementary Fig. 2a). Chromatin features indicative of a promoter state were most highly associated with annotated gene promoters, and the corresponding gene expression levels were consistent with the active or repressed state of each chromatin feature (Supplementary Fig. 2b-c).

We then assessed whether 100-CpG windows that lost the most DNA methylation in *Dnmt3a*, *Dnmt3b* and *Dnm3l* KOs were enriched at distinct chromatin features (Supplementary Fig. 1b, Fig. 2d, Supple-mentary Data 2). Highlighted on each UMAP plot are each set of *Dnmt* KO DMRs, demonstrating whether underlying chromatin signatures are linked to the activity of DNMTs to these genomic regions (Fig. 2e). This analysis identified that regions that lose DNA methylation pre-ferentially in the *Dnmt3a* KO were near inactive/active enhancers and active/bivalent promoters (Fig. 2d, e). In the *Dnmt3b* KO, H3K36me3-marked gene bodies, genic/active enhancers, and unmarked inter-genic regions (background) were affected, while in *Dnmt3l* KOs, it was almost exclusively unmarked inter-genic regions that were lost (Fig. 2d, e). Given the common enrichment for unmarked inter-genic regions, we directly compared DNA methylation between the *Dnmt3l* and *Dnmt3b* KOs, which were highly correlated (R = 0.92) (Supple-mentary Fig. 2d), suggesting that DNMT3L may act to modulate the localisation of DNMT3B to unmarked inter-genic regions in tropho-blast cells. These findings support that DNMT activities are directed by chromatin states in trophoblast development, consistent with binding preferences seen in vitro[20,23].

### DNA methylation represses germline genes in trophoblast

To determine the impact of loss of DNA methylation on gene expres-sion, we then performed ultra-low input stranded RNA-seq in E7.5 ExE (Supplementary Data 1). The transcriptomes of *Dnmt3a* KO, *Dnmt3b* KO, *Dnmt3a/b* DKO, *Dnmt3l* KO and matched WT control ExE were remarkably similar, with no distinct clustering by PCA (Fig. 3a) and correlation coefficients >0.99 compared to WTs (Fig. 3b, Supplemen-tary Fig. 3a-c). Using DESeq2 and >1.5-fold change in expression, dif-ferentially expressed genes (DEGs) were identified, with the *Dnmt3a/b* DKO being most affected with 124 DEGs (83 up-, 41 down-regulated) (Fig. 3b, Supplementary Data 3). Gene ontology analysis found that upregulated DEGs were highly enriched for germline genes, typically expressed during spermatogenesis and oogenesis, including piRNA pathway (e.g., *Piwil2, Asz1*), meiotic components (e.g., *Sycp1*) and gametic transcriptional regulators (e.g., *Dazl, Sohlh2*) (Fig. 3b, c). While there were markedly less DEGs identified in the other *Dnmt* KOs, *Dnmt3b* KO had 17 DEGs (14 up- and 3 down-regulated), of which all up-regulated were also present in the *Dnmt3a/b* DKO up-regulated DEGs

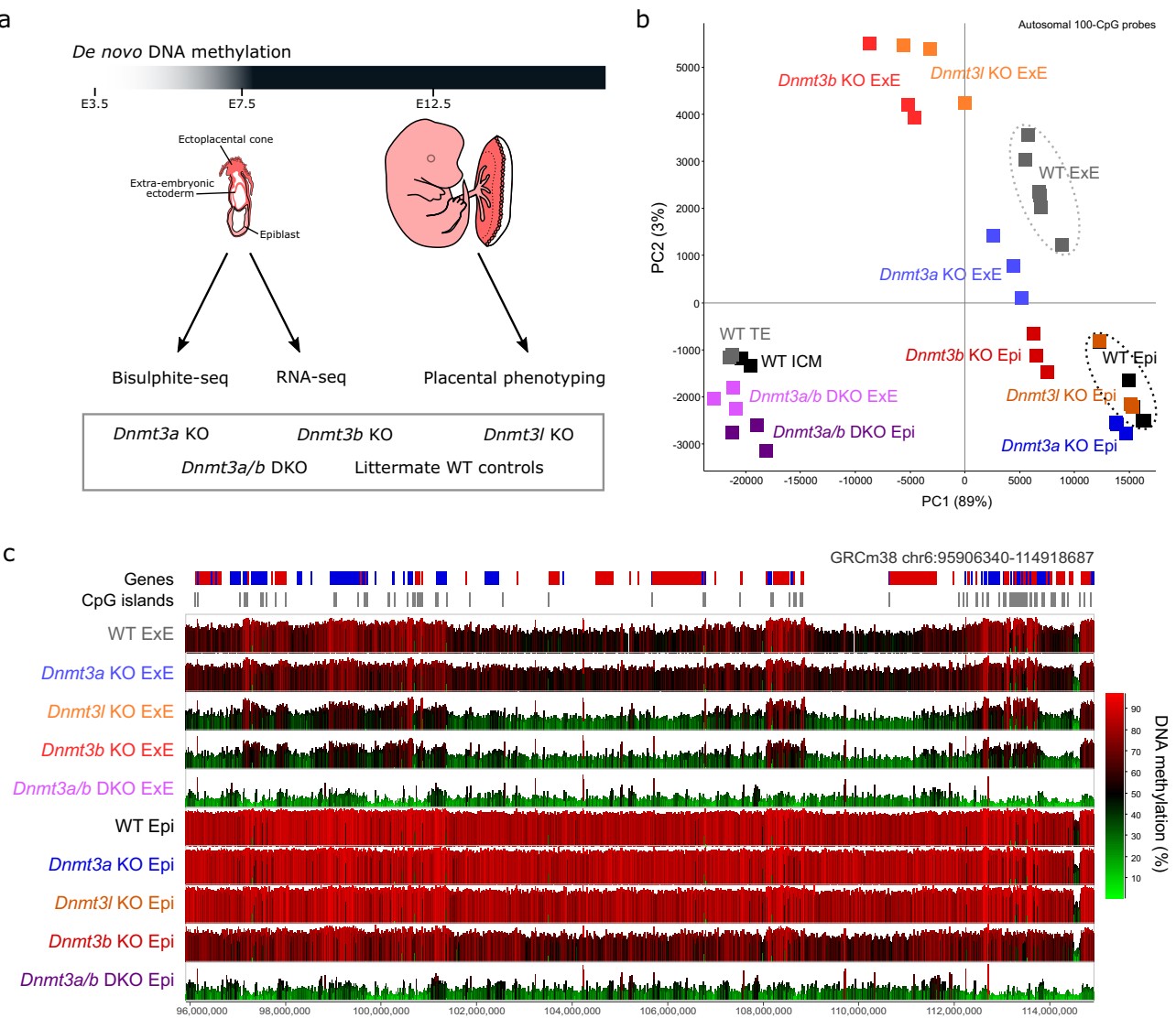

**Fig. 1 | DNMT3A, 3B and 3L mediate de novo methylation in trophoblast cells.**
**a** The schematic diagram summarises the study design. Embryos were collected at
E7.5, after the window of de novo DNA methylation, from *Dnmt3a, Dnmt3b, Dnmt3l,*
*Dnmt3a/b* DKO and wildtype (WT) controls for bisulphite- and RNA-seq. Pheno-
typing analysis was done on E12.5 placentas, when all placenta structures and cell
types are present. The E12.5 embryo drawings are adapted from Perez-Garcia, V.
*et al.* Placentation defects are highly prevalent in embryonic lethal mouse mutants.
*Nature* 555, 463-468 (2018). **b** The principle component plot shows the distribution
of DNA methylation replicates for *Dnmt3a, Dnmt3b, Dnmt3l, Dnmt3a/b* DKO and
WT E7.5 epiblast (Epi) and extra-embryonic ectoderm (ExE), in addition to publicly
available data for wildtype E3.5 inner cell mass (ICM) and trophectoderm (TE). DNA
methylation was quantitated across autosomes using 100-CpG windows with at
least 10 informative CpGs. **c** The screenshot shows the average DNA methylation
for *Dnmt3a, Dnmt3b, Dnmt3l, Dnmt3a/b* DKO and WT Epi and ExE, using 100-CpG
windows with at least 10 informative CpGs.

(Supplementary Fig. 3d) and similarly showed significant enrichment
for germline genes in gene ontology analysis (Supplementary Fig. 3e).
Hence, the gene regulatory role for de novo DNA methylation in ExE
appears to be predominantly to silence germline genes, consistent with
findings in the embryo[4,9]. We then assessed whether the de-repression
of these germline genes is correlated with the relative loss of promoter
DNA methylation. We plotted fold-change in gene expression against
the difference in promoter DNA methylation for the 83 *Dnmt3a/b* DKO
up-DEGs in each of the *Dnmt* KOs. This analysis shows that the extent of
DNA methylation loss at these gene promoters was significantly linearly
correlated with the up-regulation of gene expression (Fig. 3d).

DNA methylation is important for repression of repetitive ele-
ments in many cell contexts[4,6,7] and given the pervasive use of endo-
genous retroviruses (ERVs) in the trophoblast gene regulatory
landscape[24,25], we evaluated differential transcription of ERVs in *Dnmt*
KOs. RNA-seq libraries are only informative for a fraction of ERVs

genome-wide, so we first filtered for those with at least one same-
stranded read in at least one replicate and excluded those overlapping
an annotated gene, generating a list of 8142 'informative autosomal
ERVs.' Using the LIMMA statistical package, we identified that a small
subset of ERVs (N = 23) are significantly upregulated in the *Dnmt3a/b*
DKO (Supplementary Fig. 3f). While there was no enrichment for a
specific class of ERV (p = 0.1, Chi-square), those up-regulated ERVs
were significantly longer in sequence length than the set of informative
autosomal ERVs (p < 0.001, t-test) (Supplementary Fig. 3g-h), sugges-
tive that up-regulated ERVs are relatively intact. Upregulation of ERVs
in the *Dnmt3a/b* DKO was associated with substantial losses of DNA
methylation (median difference=53.4%), which was significantly
greater than observed in the *Dnmt3a, Dnmt3b,* and *Dnmt3l* KOs
(median difference=2.3%, 15.7% and 13.0%, respectively) (Supplemen-
tary Fig. 3i) (ANOVA p < 0.0001, *post hoc* Tukey tests p < 0.01). We
observed no significant difference in ERV de-repression between the

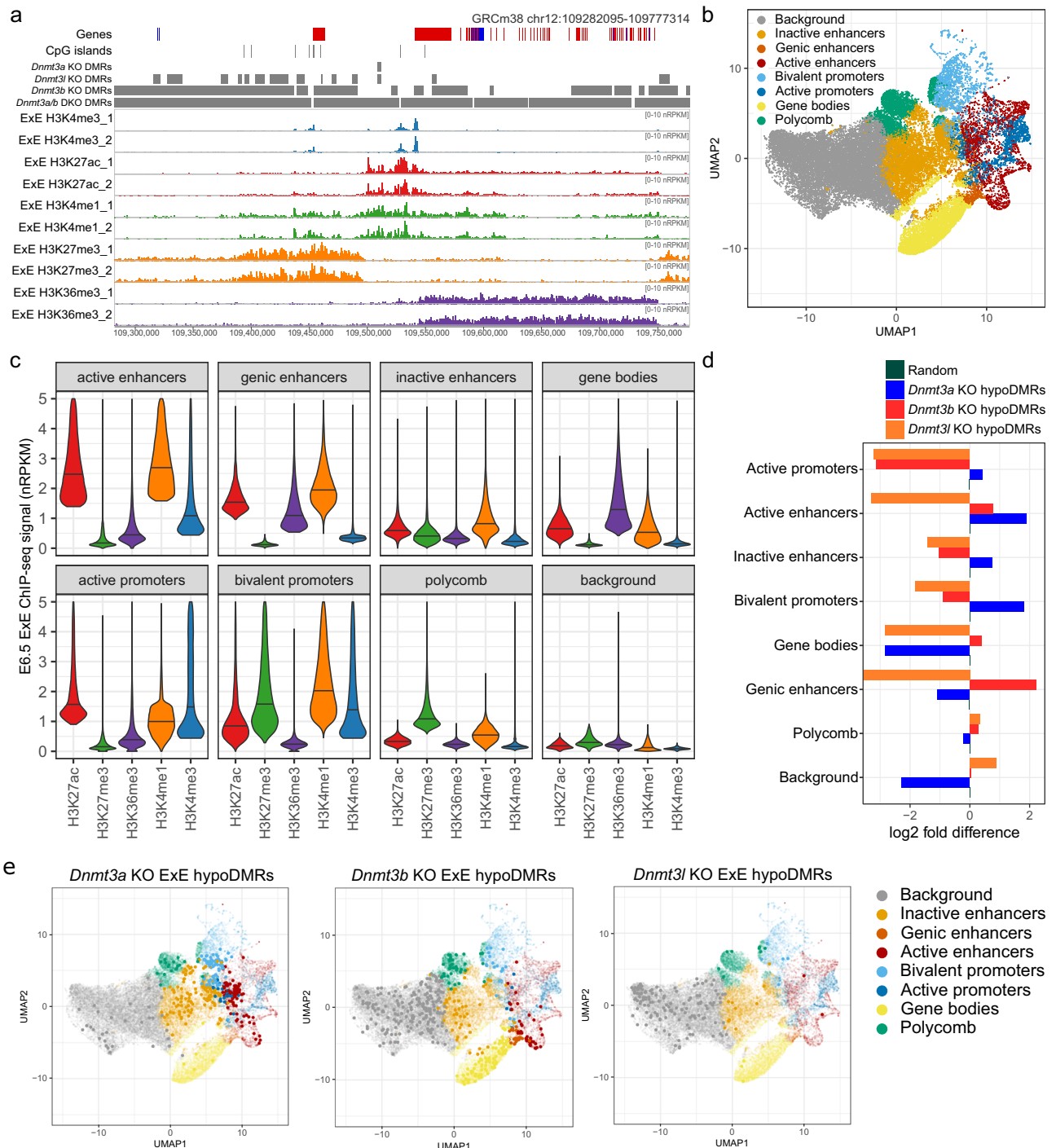

**Fig. 2 | Targeting of de novo DNMTs in ExE depends on underlying chromatin landscape. a** The genome browser shot shows ChIP-seq replicates for H3K4me3, H3K27ac, H3K4me1, H3K27me3 and H3K36me3 in E6.5 extra-embryonic ectoderm (ExE), using 1 kb running windows and enrichment normalised RPKM. Hypomethylated differentially methylated regions (DMRs) identified in the *Dnmt3a*, *Dnmt3l*, *Dnmt3b* and *Dnmt3a/b* KOs using logistic regression and >20% difference are shown as annotation tracks. **b** Eight distinct chromatin features were identified using UMAP dimensionality reduction and clustering of 100-CpG windows using histone modification abundance in ExE. **c** The bean plot shows the ChIP-seq signal for histone modifications in ExE within each defined chromatin feature. The horizontal bars show the median. **d** The bar plot shows the relative enrichment of the most hypomethylated DMRs (hypoDMRs, identified using binomial statistic) in *Dnmt3a*, *Dnmt3b* and *Dnmt3l* KO ExEs for chromatin features. Statistical comparisons are provided in Supplementary Data 2. **e** UMAP from **b**) highlighting the 100 CpG window harbouring the most hypomethylated DMRs for the respective KOs ExEs: *Dnmt3a* (N = 3,255), *Dnmt3b* (N = 3,860) and *Dnmt3l* (N = 1,281) (left to right).

*Dnmt3a*, *Dnmt3b*, and *Dnmt3l* KOs (*post hoc* Tukey test not significant for all comparisons), suggesting that substantial loss of DNA methylation is required for ERV de-repression.

Taken together, we find that de novo DNA methylation is important for repression of a remarkably small subset of the genome in E7.5 ExE, including germline genes and a set of ERVs.

**Impaired placental labyrinth formation in Dnmt3b KOs**

We next determined whether loss of DNA methylation in trophoblast cells impacted subsequent placental development. We evaluated phenotypic changes in *Dnmt3a*, *Dnmt3l* and *Dnmt3b* KOs compared to littermate WT control E12.5 placentas, a time point where the placenta is still developing but the labyrinth and junctional zone are formed.

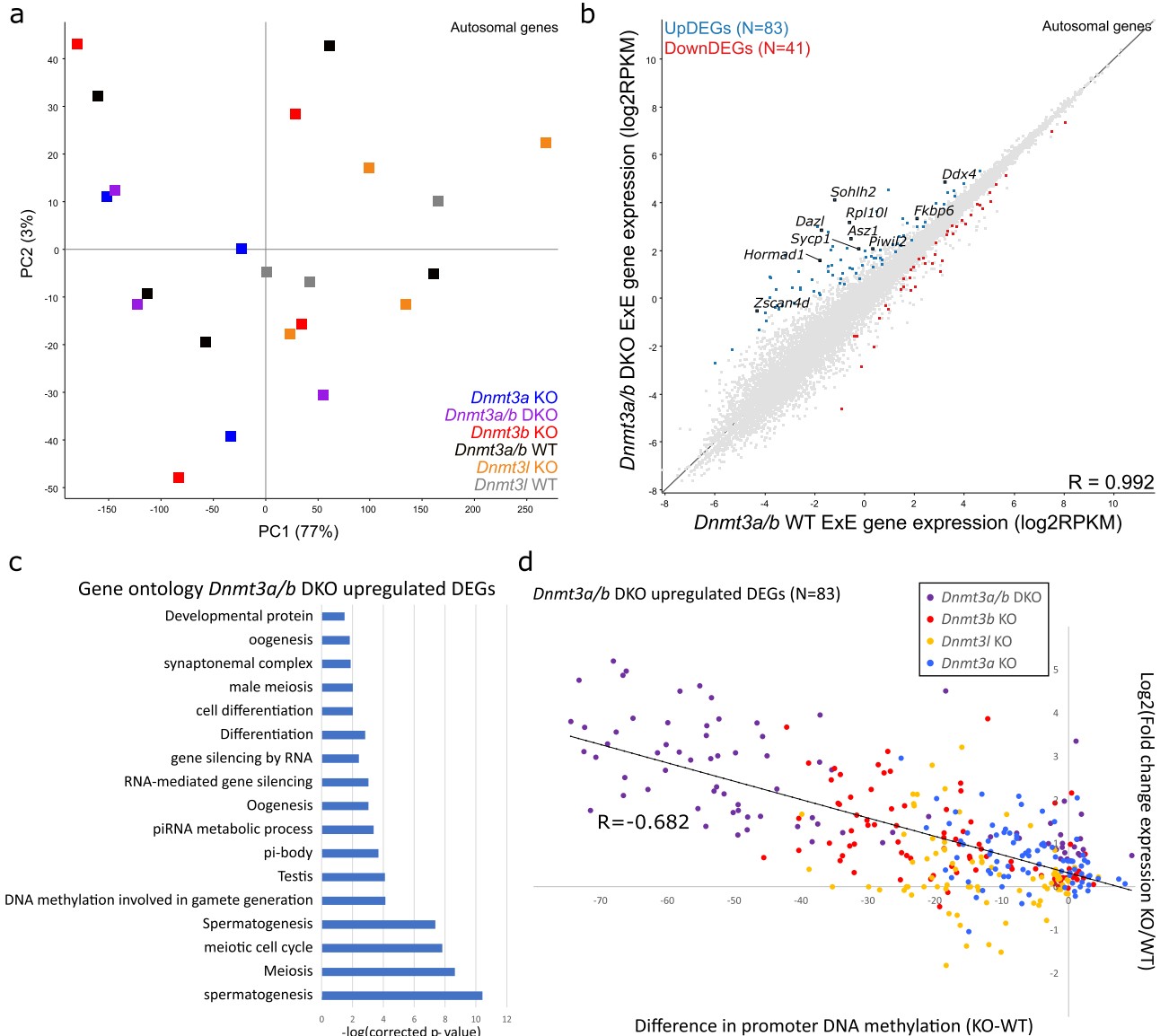

**Fig. 3 | Loss of DNA methylation in ExE results in de-repression of germline genes. a** The principle component plot shows the distribution of gene expression replicates for *Dnmt3a*, *Dnmt3b*, *Dnmt3l*, *Dnmt3a/b* DKO and wildtype (WT) extra-embryonic ectoderm (ExE). Gene expression was quantitated for autosomal genes, using log2(RPKM). **b** The scatterplot compares gene expression between *Dnmt3a/3b* DKO and WT littermate control ExE for autosomal genes. Up- and down-regulated differentially expressed genes (DEGs) are highlighted. **c** The bar plot shows significant gene ontology categories enriched among *Dnmt3a/b* DKO up-regulated genes in E7.5 ExE, using Fisher Exact Test with Benjamini-Hochberg correction for multiple comparisons. **d** The scatterplot shows the linear correlation between change in DNA methylation (KO-WT) of 100-CpG windows overlapping gene promoters with log2(fold change) in gene expression (RPKM + 0.1) in the *Dnmt3a/b* DKO, *Dnmt3b* KO, *Dnmt3a* KO and *Dnmt3l* KO ($p < 0.0001$, linear correlation test statistic).

*Dnmt3a/b* DKOs were not assessed, as these embryos die by E10.5 and hence never form a functional placenta. We first evaluated placental morphology using haematoxylin and eosin staining, with no dramatic effects or apparent differences in size seen in *Dnmt3a* or *Dnmt3l* KOs compared to WTs (Supplementary Fig. 4a-b). However, *Dnmt3b* KO placentas were smaller, which we linked to an underdeveloped labyrinth zone (Fig. 4a, Supplementary Fig. 4c), the main nutrient and gas-exchange surface in the mouse placenta. Immunohistochemistry for E-Cadherin (CDH1) confirmed impaired development of the syncytiotrophoblast (SynT) within the labyrinth of *Dnmt3b* KO placentas (Supplementary Fig. 4d). To further explore the labyrinth defect, immunofluorescence analysis for monocarboxylate transporters 1 (MCT1) and 4 (MCT4) was used to demarcate the SynTI and SynTII layers, respectively, that separate the maternal and foetal blood circulation. We observed a substantial reduction in the SynT surface

available for nutrient transport in *Dnmt3b* KO placentas, with the SynTII layer being most affected (Fig. 4b). The mis-formation of the maternal-foetal interface of the placental labyrinth was further corroborated by isolectin B4 staining, highlighting the overall foetal vasculature organisation, which confirmed that *Dnmt3b* KO placentas have an impaired vascularisation (Fig. 4c). The defects in SynT development and vascularisation indicate impaired foetal-maternal exchange[26] in *Dnmt3b* KO placentas, likely compromising foetal growth and survival.

To investigate the placental phenotype of *Dnmt3b* KO at the molecular level, we generated single-nuclei RNA sequencing data from E12.5 placentas (Supplementary Fig. 5a-b). Using published gene signatures for mouse placental cell types[27], we were able to identify all placental cell types in both the *Dnmt3b* KO and WT (Fig. 4d, Supplementary Fig. 5c-d). Consistent with the reduced size of the labyrinth

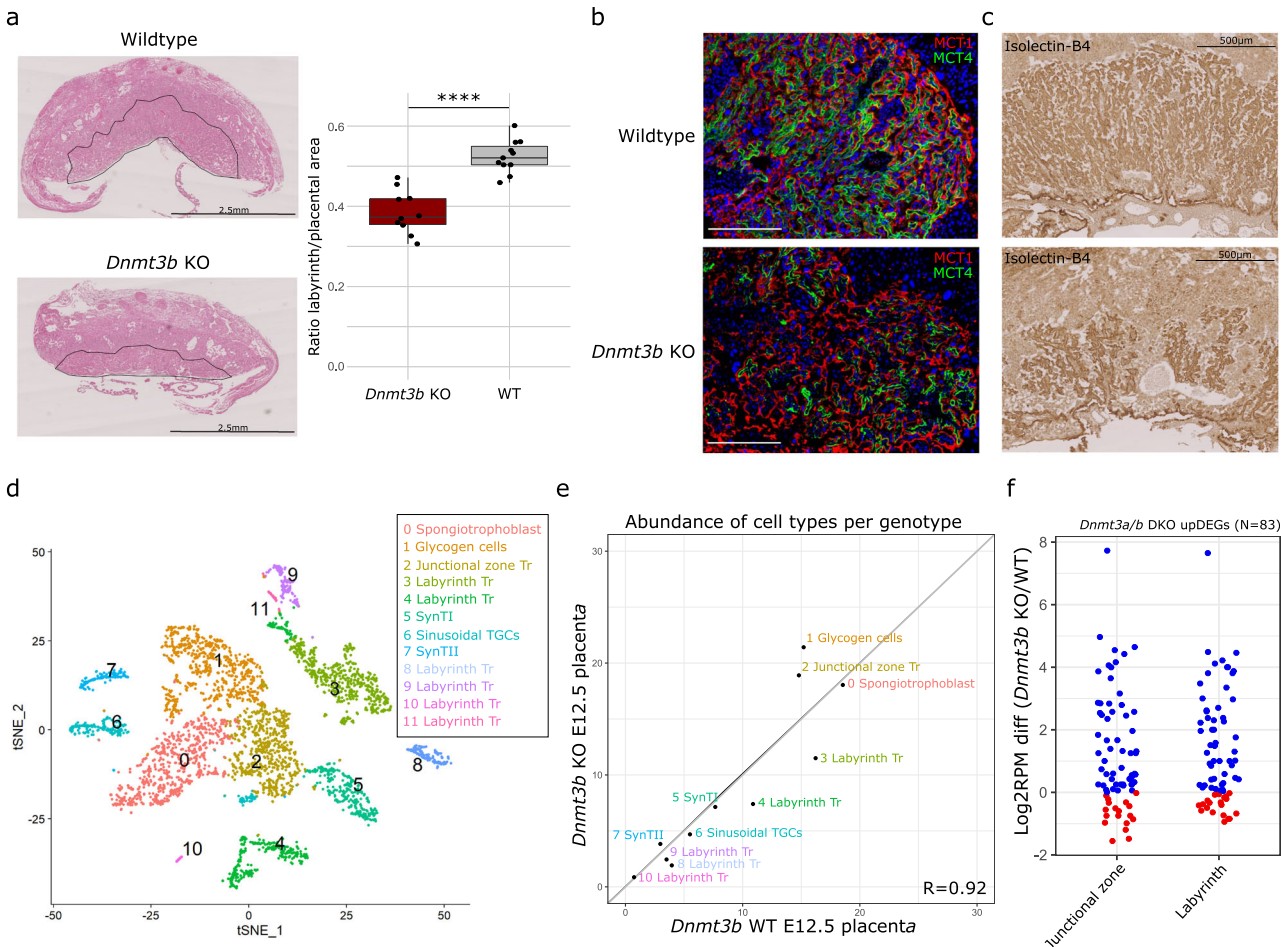

**Fig. 4 | *Dnmt3b* KOs show impaired formation of the placental labyrinth.**
**a** Representative haematoxylin and eosin staining of E12.5 placentas from a *Dnmt3b* KO and littermate WT control (left). The dashed outlined region corresponds to the placental labyrinth zone. The boxplot show the ratio of labyrinth to placental area in *Dnmt3b* KO ($N = 11$) and WT ($N = 10$) E12.5 placentas (two-tailed t-test ****$p = 0.000002$) (right). The boxplot centre line is the median, with box limits showing the upper and lower quartiles and whiskers as 1.5x interquartile range. Individual data points are shown as dots. **b** Immunofluorescence for MCT1 and MCT4, which stain for syncytiotrophoblast layer I (SynTI) and SynTII that separate maternal and foetal circulation, respectively, of E12.5 placentas from a *Dnmt3b* KO and littermate WT control. White scale bar = 100 μm. **c** Representative isolectin BSI-B4 immunohistochemistry of E12.5 placentas from a *Dnmt3b* KO and littermate WT control, highlighting the matrix surrounding foetal vessels. **d** t-distributed stochastic neighbour embedding (t-SNE) plots of placental cell types characterised by single-nuclei RNA-sequencing data from E12.5 placentas from *Dnmt3b* KO and littermate WT control. Tr=Trophoblast; TGCs=trophoblast giant cells. **e** The scatterplot shows the relative abundance of cell types within the *Dnmt3b* KO and littermate WT control E12.5 placentas ($p < 0.0001$, linear correlation test statistic). **f** The dot plot shows the difference in expression between *Dnmt3b* KO and WT for genes upregulated in the *Dnmt3a/b* DKO ExE ($N = 83$). Log2(RPM) was calculated for each gene among cells within the junctional zone and labyrinth from the single nuclei RNA-seq data from E12.5 placenta. Genes that are up- or downregulated in the *Dnmt3b* KO compared to WT are shown in blue and red, respectively. The proportion of genes up-regulated in *Dnmt3b* KO was significant in the junctional zone ($p = 0.002$) and labyrinth ($p = 0.006$) (two-tailed Fisher's Exact test).

compartment seen in *Dnmt3b* KO placentas, we observed a subtle decrease in the abundance of labyrinth trophoblasts, with a concomitant increase in junctional zone trophoblast and glycogen cells (Fig. 4e). However, the cell composition of *Dnmt3b* KO and WT placentas was highly correlated ($R = 0.92$) (Fig. 4e).

We then assessed whether there was evidence for dysregulated gene expression in *Dnmt3b* KO SynT cells compared to WT (Supplementary Data 4). Gene ontology analysis of upregulated DEGs ($N = 36$) showed similar results to those seen for *Dnmt3a/b* DKO and *Dnmt3b* KO E7.5 ExE, with a significant de-repression of germline genes in E12.5 SynT cells (Supplementary Fig. 5e). Notably, one significant gene ontology category was enriched among down-regulated DEGs ($N = 18$) in *Dnmt3b* KO SynT cells (Supplementary Data 4), the tissue category "placenta," which included several genes that are highly expressed placenta (e.g., *Cts3*, *Ascl4*, *Fn1*, *Ceacam14*), but no specific pathway or processes was identified (Supplementary Fig. 5e). To determine whether de-repression of germline genes was specific to SynT cells or was

observed across placental compartments, we measured the expression of the 83 upregulated DEGs identified in the *Dnmt3a/b* DKO ExE in E12.5 junctional zone and labyrinth, which both showed significantly upregulated expression in the *Dnmt3b* KO (Fig. 4f, Supplementary Fig. 5f). These data suggest that while differentiation does not appear to be affected in *Dnmt3b* KO placentas, there is widespread de-repression of germline genes and compromised expression levels of some placental genes.

In summary, loss of DNMT3B results in a severe impairment in the formation of the placental labyrinth, but this effect does not appear to be underpinned by a failure of trophoblast differentiation or cell identity.

**Dnmt3b KO mid-gestation embryonic lethality is rescued with a functional placenta**
The formation of the placental labyrinth is initiated when the allantois, a projection of extra-embryonic mesodermal cells derived from the

epiblast, fuses with the chorionic ectoderm at E8.5[26]. Predominantly thought to be mediated through WNT, FGF, BMP and HGF signalling, the invading mesoderm induces branching and differentiation of the chorionic trophoblast cells to form the two SynT layers[28]. Therefore, defects in labyrinth formation may either originate from impaired allantoic signals, or from deficiencies intrinsic to the trophoblast compartment itself. To distinguish between these possibilities, we generated conditional *Dnmt3b* KO (cKO) conceptuses using *Sox2*-Cre, a transgenic mouse line that uses the endogenous *Sox2* promoter to confer epiblast-specific Cre expression, which is widely used for the generation of embryo-specific KOs[29] (Supplementary Fig. 6a-b). Consequently, the *Dnmt3b* cKO results in conditional deletion of *Dnmt3b* in all embryonic lineages before post-implantation de novo DNA methylation (Fig. 5a). We confirmed the specificity of the cKO with PBAT of *Dnmt3b* cKO E7.5 epiblast and ExE and PCR genotyping of the *Dnmt3b* allele in E8.5 whole embryos and placentas (Supplementary Fig. 6c-d). Immunofluorescence analysis of MCT1 and MCT4 in *Dnmt3b* cKO and WT E12.5 placentas showed that the labyrinth defects seen in the *Dnmt3b* KO placentas were fully rescued in the *Dnmt3b* cKO (Fig. 5b, Supplementary Fig. 6e). We performed single-nuclei RNA-sequencing on *Dnmt3b* cKO and WT E12.5 placentas, demonstrating that gene expression profiles in the *Dnmt3b* cKO resemble WT (Fig. 5c). Therefore, we conclude that DNMT3B is critical in trophoblast cells,

rather than the allantois, in mediating appropriate formation of the placental labyrinth.

We then assessed whether the presence of a functional placenta in the *Dnmt3b* cKO could rescue embryo survival. At E12.5, *Dnmt3b* cKO embryos showed a remarkably improved morphology compared to *Dnmt3b* KO embryos (Fig. 5d). *Dnmt3b* KO embryos die at approximately E13.5[7], and consistently we recovered no *Dnmt3b* KO embryos in late gestation; yet, *Dnmt3b* cKO embryos survived until just before birth at E18.5, at expected Mendelian ratios (Fig. 5e, Supplementary Fig. 6f-g). Immunofluorescence of E18.5 placentas confirmed phenotypically normal placental labyrinth development in *Dnmt3b* cKOs throughout pregnancy (Supplementary Fig. 6h). Nevertheless, at E18.5, *Dnmt3b* cKO embryos showed significantly reduced foetal weight and no *Dnmt3b* cKO embryos survived to postnatal day 10 (Fig. 5e-f, Supplementary Fig. 6g). Hence, DNMT3B has a late-gestational role in the embryo proper impacting perinatal survival; the phenotype underpinning this perinatal lethality has not been studied here. Our findings demonstrate that impaired formation of the maternal-foetal interface in the placental labyrinth underlies the embryonic lethality observed in *Dnmt3b* KOs.

## Discussion

In this study, we demonstrate that DNMT3A, DNMT3B and DNMT3L are all integral for the establishment of the placental methylome

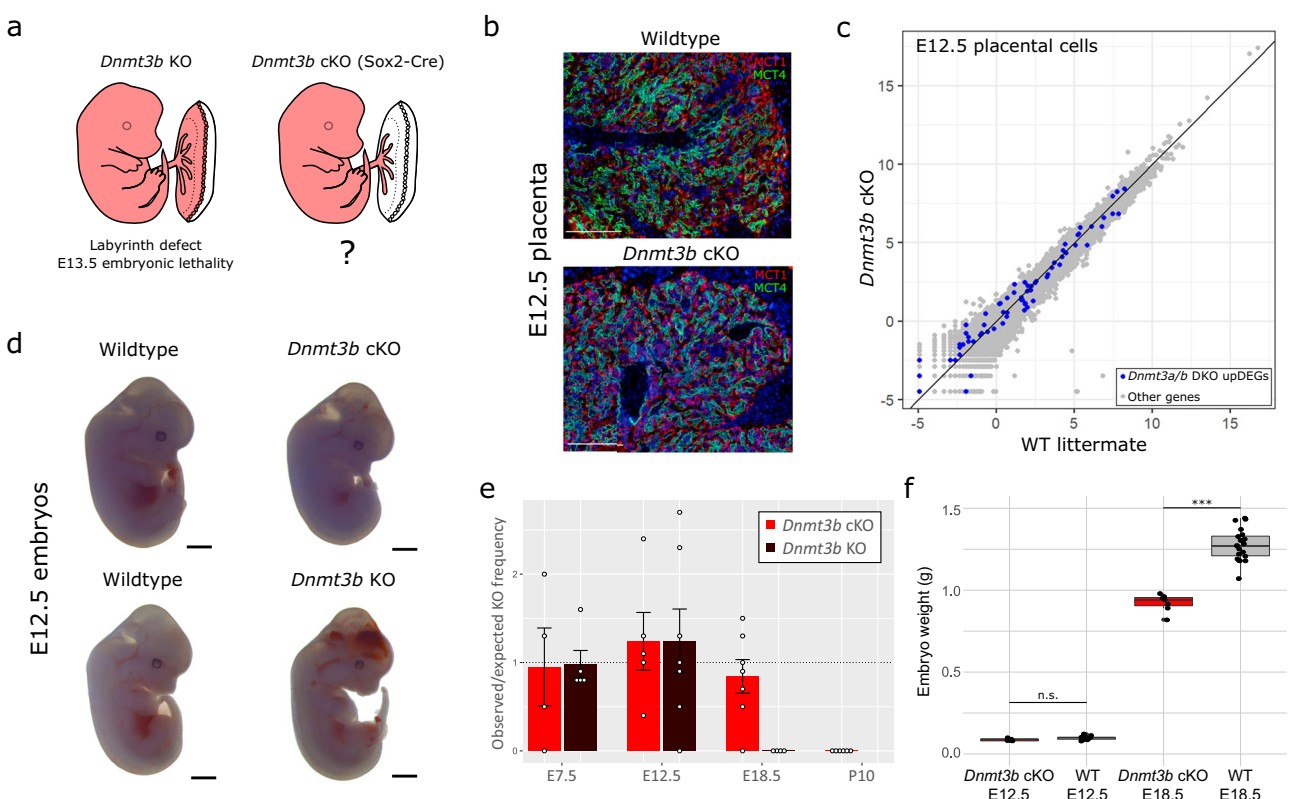

**Fig. 5 | *Dnmt3b* cKOs show rescued placental labyrinth phenotype and consequently survive to birth. a** The schematic diagram shows the lineages containing *Dnmt3b* deletion in the *Dnmt3b* KO compared to the *Dnmt3b* cKO, using Sox2-Cre. This schematic is adapted from Perez-Garcia, V. *et al*. Placentation defects are highly prevalent in embryonic lethal mouse mutants. *Nature* 555, 463-468 (2018). **b** Immunofluorescence for MCT1 (SynTI) and MCT4 (SynTII) in E12.5 placentas from a *Dnmt3b* cKO and littermate WT control. White scale bar = 100 μm. **c** The scatterplot shows gene expression, evaluated by single-nuclei RNA-sequencing in E12.5 placental cells, in *Dnmt3b* cKO, using Sox2-Cre, compared to WT littermate control. Cells were pseudo-bulked to generate log2(RPM) values for *Dnmt3a/b* DKO upregulated DEGs and other genes. **d** Images show E12.5 embryos from *Dnmt3b* KO, *Dnmt3b* cKO and littermate WT

controls. Black scale bar = 1 cm. **e** The bar plot shows the mean observed/ expected ratios of KO or cKO embryos at E7.5, E12.5, E18.5 and postnatal day (P) 10 in the *Dnmt3b* KO and cKO crosses, respectively. The number of litters evaluated for the *Dnmt3b* KO and cKO, respectively, at: E7.5 (*N* = 5 and 4), E12.5 (*N* = 7 and 5), E18.5 (N = 4 and 7), and P10 (N = 0 and 6). Error bars show standard error of the mean. **f** E12.5 and E18.5 foetal weights were compared between *Dnmt3b* cKO (*N* = 7 and 7, respectively) and littermate WT (WT and heterozygous) controls (*N* = 21 and 21, respectively) from four pregnancies for each time point (two-tailed t-test with Bonferroni correction for multiple comparisons, n.s. = not significant, ***p = 9.1E-10). The boxplot centre line is the median, with box limits showing the upper and lower quartiles and whiskers as 1.5x interquartile range. Individual data points are shown the dots.

during post-implantation development. We also show that each DNMT preferentially targets a set of highly specific genomic features demarcated by characteristic chromatin states. Importantly, de novo DNA methylation plays a critical role in healthy placental development, with loss of DNMT3B resulting in impaired formation of the placental labyrinth causing mid-gestation embryonic lethality. Despite the striking phenotype of *Dnmt3b* knockout placentas, we observed no impairment of specification or differentiation of placental cell types. Rather, we find that the prominent transcriptional change in both undifferentiated and differentiated trophoblast cells was up-regulation of germline genes normally expressed in oogenesis and spermatogenesis. This finding mirrors observations made in embryonic cell types[4,8,9], and supports the notion that one of the primary roles of DNA methylation is to silence germline genes[30].

The DNMT proteins contain three histone-interacting domains: DNMT3A contains UDR, ADD and PWWP domains, DNMT3B contains ADD and PWWP domains and DNMT3L contains an ADD domain. Furthermore, the DNMTs can form hetero-dimers and oligomers, making the patterns of recruitment in biological contexts challenging to disentangle[31]. Indeed, studies investigating the targeting of de novo DNMTs in the embryo observed relatively uniform kinetics throughout the genome[4], with later HiC studies showing that A compartments methylate before B compartments[32]. However, during epigenetic programming in trophoblast cells, we observe distinct genomic targeting preferences for DNMT3A, DNMT3B and DNMT3L activity, linked to histone modifications. DNMT3B-targeted sites show enrichment for H3K36me3-marked gene bodies in trophoblast cells, consistent with its localisation in vitro via its PWWP domain[20]. Conversely, DNMT3L-dependent regions appear to be largely devoid of H3K4me and H3K36me3, consistent with ADD domain-directed localisation of DNMT3L to unmethylated H3K4[19]. DNMT3L enhances the catalytic activity of de novo DNMTs[33,34], thus one could speculate that its trophoblast-specific function may be necessary due to the lower expression levels of *Dnmt3a* and *Dnmt3b* compared to epiblast, enabling recruitment of DNMTs to low-affinity inter-genic sites.

While DNMT3A can bind to unmethylated H3K4 and H3K36me2/3 through its ADD and PWWP domains, respectively, we find that DNMT3A-targeted sites are most prominent at the boundaries of active and bivalent domains in trophoblast cells. Recent in vitro work has shown that the localisation of DNMT3A to bivalent regions of the genome is underpinned by an interaction of its N-terminal UDR domain with H2AK119ub in the absence of PWWP recruitment to H3K36me2 domains[35]. This mechanism may be relevant in trophoblast cells, as it has been reported that H3K27me3-marked CpG islands, which often co-localises with H2AK119ub, are atypically methylated in ExE[16]. Interestingly, a recent study reported almost wildtype levels of DNA methylation and gene expression in E11.5 embryos with a catalytically-inactivated DNMT3B[36], suggesting that catalytically-inactive DNMT3B can heterodimerise with DNMT3A to restore DNA methylation patterning. While placental cells were not assessed in this study, the rescue of embryonic lethality in this model suggests that similar mechanisms may occur in trophoblast cells. Future studies will be valuable in understanding whether these various mechanisms of DNMT3A-recruitment differ across cell types. To date, the link between genome-wide de novo DNMT activity and chromatin landscape in vivo has only been well-documented in the context of epigenetic programming in gametogenesis[37–39]. Our findings reveal that similar mechanisms may influence DNA methylation programming during trophoblast development, with chromatin-mediated recruitment of DNMTs likely explaining at least some of the unique features of the placental methylome.

The accumulation of DNA methylation in a lineage-specific manner in E7.5 epiblast and ExE is consistent with its suggested role in re-enforcing transcriptional repression associated with lineage commitment[40]. However, we do not observe precocious activation of development genes in *Dnmt* KO ExE, nor impaired differentiation or tissue composition in the *Dnmt3b* KO placenta when assessed by single-nuclei RNA-sequencing. This finding was surprising given the markedly impaired formation of the SynTII layer seen in *Dnmt3b* KO placental sections. SynTII cells were defined in single nuclei analysis using published markers, including *Gcm1*, *Synb*, *Gcgr* and *Vegfa*[27], while the identification of SynTII cells by immunofluorescence was done using the widely adopted approach of staining for lactate proton-coupled monocarboxylate transporter protein MCT4 (*Slc16a3*)[41,42]. Transcription factor, GCM1, becomes expressed in the chorionic trophoblast at the branchpoints of the allantoic mesoderm and embryos that lack *Gcm1* fail to initiate differentiation and formation of the placental labyrinth[28]. MCT4 is a lactate transporter expressed in differentiated SynTII cells and is essential in its function in maternal-foetal exchange[41]. Thus, we interpret our findings to show that specification of SynTII can occur in *Dnmt3b* KO placentas, but that the morphogenesis and function is severely compromised, resulting in a failure to support foetal development beyond E13.5. Our data suggests that the de-repression of germline genes is interfering with these processes in SynTII formation and function. However, an important limitation of single cell RNA-seq is its inability to detect low and intermediately expressed genes, thus while germline gene de-repression is the prominent differential signature of *Dnmt3b* KO placental cells, further relatively subtle changes in the gene regulatory landscape will likely be overlooked. Intriguingly, impairment in branching morphogenesis and transcriptional de-repression of germline genes has also been observed in the lungs of *Dnmt1* cKO mice[43], suggesting that failure to repress germline genes may interfere with branching morphogenesis processes across developmental contexts. Future studies will be valuable in gaining further insights into the molecular mechanisms underpinning these phenotypes.

DNA methylation is studied extensively in the context of human pregnancy for association with placental insufficiency and pregnancy complications, such as preeclampsia, growth restriction and preterm birth[44]. Yet, there remain few examples of a functional role for DNA methylation in placentation outside the context of genomic imprinting and X chromosome inactivation. Our study reveals that germline genes and a subset of endogenous retroviruses are susceptible to de-repression upon loss of DNA methylation. Further, we identify formation of the placental labyrinth as a developmental process critically reliant on correct establishment of DNA methylation in trophoblast cells. Pregnancy complications affect as many as one in four women and maternal health during pregnancy has lasting impacts on the lifelong health of offspring. Our findings provide invaluable context for studies investigating environmental exposures in pregnancy and developmental origins of health and disease and will help direct future studies of epigenetic mechanisms underpinning human placental development.

## Methods

### Sample collections

The use of animals in this study was performed in accordance with the European regulation in Animals (Scientific Procedures) Act 1986 with all protocols approved by the Animal Welfare and Ethical Review Body at the Babraham Institute under licenses issued by the Home Office (UK).

Natural timed matings were used to collect embryos on embryonic days 6.5 (E6.5), E7.5, E12.5, and E18.5. We used published mouse models for *Dnmt3l*[5], *Dnmt3a*[7], and *Dnmt3b*[7], and *Sox2*-Cre[29], which have been bred into the C57BL6/Babr background for this study. To generate KO and wildtype embryos, heterozygous females were bred to heterozygous males for *Dnmt3l*, *Dnmt3a*, *Dnmt3b* and *Dnmt3a/3b* strains. To generate *Dnmt3b* cKO embryos, homozygous *Dnmt3b* floxed females (*Dnmt3b* fl/fl) were bred to heterozygous *Dnmt3b* floxed males carrying the Sox2-Cre (*Dnmt3b* fl/+, Sox2-Cre +ve). For all

strains used in this study, heterozygous animals are phenotypically normal.

At E6.5 and E7.5, the epiblast (Epi), extra-embryonic ectoderm (ExE) and ectoplacental cone (EPC) for each embryo were manually separated. Single E7.5 epiblast and ExE samples were individually frozen in 10 μL of buffer RLT Plus (Qiagen) for RNA-seq or PBAT; the corresponding EPC was used for genotyping. For ChIP-seq, pools of C57BL6/CAST E6.5 ExE (N = 8) samples were washed in PBS and flash frozen in 10 μL of nuclear lysis buffer (Sigma), matching the strain and collection approach for incorporated published datasets[18]. For placental phenotyping, E12.5 (and E18.5 for the *Dnmt3b* cKO) placentas were collected and fixed overnight in 4% paraformaldehyde at 4 °C, washed with PBS and stored at 4 °C until paraffin embedded. For E12.5 and E18.5 embryos, a tail clip was used for genotyping. For all molecular and phenotyping experiments, wherever possible, matched littermate wildtype controls were used for comparisons to KOs.

EPC and tail clip samples were lysed in 50 mM Tris pH8.5, 1 mM EDTA, 0.5% Tween-20, and 0.2 μg/μl proteinase K at 55 °C for 3 h, followed by 95 °C 10 min incubation. MyTaq Red Mix (Meridian Bioscience) was used to amplify 1 μL DNA in a 20 μL reaction using a 60 °C annealing temperature and 35 amplification cycles. The following primers were used for each locus: *Dnmt3a*-F CTGTGGCATCT-CAGGGTGATGAGCA, R1 GCAAACAGACCCAACATGGAACCCT, R2 TGAGTGGTGAGGCCC-AGCTTATCGA (Approximate sizes: WT-400bp, KO-250bp); *Dnmt3b*-F1 GAACTTGGTCTGCAGGACGAT-CGCT, F2 AGAGCACTGCACCACTACTGCTGGA, R CAGGTCAGACCTCTCTGG TGACAAG (Approximate sizes: WT-270bp, floxed-400bp, KO-700bp); *Dnmt3l*-F1 GGTCCTTAGGGGTTCTGGAC, F2 GTTGGAGGA-TTGGGAA-GACA, R1 TAGCTACCCGTGGCCAATAC, R2 CCATGGCATTGATCCT CTCT (Approximate sizes: WT-250bp, KO-170bp); Sox2-Cre-F GCA-GAACCTGAAGATGTTCGCGAT, R AGGTATCTCTGACCAGAGTC-ATCC (Approximate size: 700 bp). PCR products were run on a 2% agarose gel and imaged using a Gel Doc XR + System (Bio-Rad).

## Immunohistochemistry and immunofluorescence

For histological analysis, at least 4 mutant and 4 wild-type or heterozygous placentas were collected from at least two independent litters (with mutant and wild-type placentas recovered from each litter) for *Dnmt3a* KO, *Dnmt3b* KO, *Dnmt3l* KO and *Dnmt3b* cKO. Placentas from each litter were processed for routine paraffin histology and embedded side-by-side within a paraffin block. No other randomization is applicable for this study. In all cases, tissue appearance and cellular architecture of the placentas was analysed to confirm they were in viable condition. Consecutive 10μm sections were produced. A series of sections per block was processed for haematoxylin and eosin (H&E) staining, using a standard protocol[42]. Sections through the sagittal midline were chosen for imaging, indicated at E12.5 and E18.5 by the site of insertion of the umbilical cord. Slides were scanned on a Nanozoomer (Hamamatsu). Phenotypes of placentas were assessed for each strain, blinded and recorded by at least two independent investigators.

For immunostaining, sections were deparaffinised in xylene and processed through an ethanol series to PBS. Antigen retrieval was performed by boiling in 1 mM EDTA pH 7.2, 0.05% Tween-20 or in 10 mM Na-citrate pH 6.0 buffer followed by blocking in PBS, 0.5% BSA, 0.1% Tween-20. Antibodies used were anti-Cdh1 (1:100; BD Biosciences 610181), anti-MCT1 (1:100; Merck Millipore AB1286I), anti-MCT4 (1:100; Merck Millipore AB3314P) and biotin-conjugated isolectin from *Bandeiraea simplicifolia* BSI-B4 (1:100; Sigma L2140). Primary antibodies were detected with appropriate fluorescence or horseradish peroxidase-conjugated secondary antibodies; BSI-B4 was detected with horseradish peroxidase-conjugated streptavidin (1:400 Alexa594 (Goat anti-chicken) Invitrogen Cat. A110442, 1:400 Alexa488 (Donkey anti-rabbit) Invitrogen Cat. A21206, 1:200 Anti-Mouse IgG (H + L)-HRP (goat polyclonal) Cat. 170–6516 Bio-Rad, 1:100 Horseradish

peroxidase-conjugated Streptavidin, Vector Laboratories Cat. SA5004). Nuclei were counterstained with haematoxylin or DAPI. For all immunostainings, samples were repeated at least twice in independent experiments. Images were taken with an Olympus BX61 epifluorescence microscope and/or Arperio scanner (Leica). The signal for MCT4 and MCT1 staining was quantified using ImageJ and Student's t-test analysis was performed to calculate statistical significance of the staining signal differences (p < 0.05) using GraphPad Prism 9.

## RNA-seq library preparation

Ultra-low input RNA-sequencing libraries were generated as previously described[18], with the adaptation of using only a single round of oligo(dT)25 capture of mRNA to improve yields. In brief, samples were homogenized in 400 μL of TRIzol (Invitrogen) and phase separated with 80 μL of chloroform:isoamyl alcohol (Sigma), centrifuging at 4 °C for 15 min. The aqueous phase was combined with 1 μL GlycoBlue and 300 μL of cold isopropanol. The RNA was pelleted by centrifugation for 10 min at 4 °C, washed once with 75% ethanol, dried and then resuspended in 5 μL of ultra-pure water. To each RNA sample, 20 μL of lysis/binding buffer was immediately added, proceeding directly to oligo(dT)25 capture of mRNA, using Dynabeads mRNA DIRECT kit (Life Technologies), using previously described adaptations for low starting material[18]. Using the SMARTer Stranded RNA-seq kit (Clontech), mRNA libraries were prepared as per the manufacturers' instructions with 15 amplification cycles. Libraries were quantified using the High DNA Sensitivity Bioanalyzer 2500 (Agilent) and Illumina library quantification (KAPA). Libraries were multiplexed and sequenced using 50 bp single-end on the Illumina HiSeq 2500 RapidRun. Four replicates were excluded due to poor amplification and consequently high duplication, including 2 KO and 2 WT replicates. In total, the number of replicates analysed per group were *Dnmt3l* KO (N = 4), *Dnmt3l* WT (N = 3), *Dnmt3a/b* DKO (N = 3), *Dnmt3b* KO (N = 4), *Dnmt3a* KO (N = 3), *Dnmt3a/b* WT (N = 5).

## PBAT library preparation

PBAT libraries were generated as previously described[45]. In brief, cells were lysed with 0.5% SDS at 37 °C for 1 h and bisulphite converted using the Imprint DNA Modification kit (Sigma). Bisulphite-converted DNA was purified using the EZ DNA Methylation Direct kit, as directed (Zymo Research). Using a biotin-conjugated adaptor containing standard Illumina adaptor sequences and 9 bp of random sequences (9 N), first-strand synthesis was performed using Klenow Exo- enzyme (New England Biolabs). Following exonuclease (New England Biolabs) treatment and binding to Dynabeads M-280 Streptavidin beads (Thermo Fisher Scientific), second-strand synthesis was performed. Libraries were amplified and indexed using 10 PCR cycles with Phusion High-Fidelity DNA polymerase (New England Biolabs). Libraries were multiplexed for 75-bp paired-end sequencing on an Illumina NextSeq500. Three replicates of Epi and ExE were sequenced and analysed per group, as detailed in Supplementary Data 1.

## ChIP-seq library preparation

Ultra-low input ChIP-seq was performed as previously described[45], incorporating several adaptions from the original protocol[46]. In brief, replicates of ExE samples were permeabilized in nuclei EZ lysis buffer (Sigma) with 0.1% Triton-X-100/0.1% deoxycholate. Chromatin digestion was completed with 200 U of micrococcal nuclease (New England Biolabs) at 21 °C for 7.5 min. Chromatin was then precleared in complete immunoprecipitation buffer with Protein A/G beads, for 2 h rotating at 4 °C. Chromatin was then divided into three aliquots: one for each immunoprecipitation and one 10% input control. Antibodies for H3K4me1 (250 ng, Active Motif, 39298) or H3K27ac (125 ng, Abcam, ab4729) were bound to Protein A/G beads in complete immunoprecipitation buffer for 3 h rotating at 4 °C. Chromatin was added to antibody-bound beads and rotated overnight at 4 °C. Chromatin-

bound beads were washed with two low-salt washes and one high-salt wash, followed by DNA elution at 65 °C for 90 mins. Eluted DNA was purified with Sera-Mag carboxylate-modified Magnetic SpeedBeads (Fisher Scientific) at a 1.8:1 ratio. Library preparation and indexing was performed using the MicroPlex Library Preparation kit v2 (Diagenode), as per the manufacturer's instructions. Libraries were multiplexed for 75-bp paired-end sequencing on an Illumina NextSeq500.

## Single-nuclei RNA-seq library preparation

Single nuclei were isolated as described in the 'Frankenstein' protocol, available through 10X Genomics (https://www.10xgenomics.com/resources/customer-developed-protocols) and published adaptations for mouse placenta[27]. Following collection and genotyping, *Dnmt3b* KO and WT E12.5 placentas or *Dnmt3b* cKO and WT E12.5 placentas were processed in parallel immediately for single nuclei isolation and 10X Genomics RNA-seq preparation. Placentas were first chopped with a razor blade in 1 mL nuclei EZ lysis buffer (Sigma) for 2 mins, followed by homogenisation in a Dounce homogenizer. Nuclei were filtered through a 30 μm filter, and then pelleted by centrifugation at 500 g for 5 mins at 4 °C. Nuclei were resuspended in fresh nuclei EZ lysis buffer, pelleted and then resuspended in nuclei wash and resuspension buffer (1% BSA, 0.2U/μL RNase Inhibitor, PBS). Nuclei were gently resuspended, pelleted and resuspended in fresh nuclei wash and resuspension buffer. Nuclei were passed again through a 30 μm filter, before visual inspection of nuclei with a haemocytometer. Nuclei were then stained with DAPI (adding 1 μL of 1 mg/mL DAPI) before a total of 10,000 nuclei per sample were flow sorted using the BD FACSAria Fusion sorter with 70 μm nozzle into RT reaction mix (10X Genomics). Samples were immediately processed to cDNA using the Chromium Controller (10X Genomics), followed by 3' single-nuclei RNA-seq library preparation (10X Genomics). Libraries were multiplexed for sequencing on an Illumina NextSeq500.

## Public datasets

Publicly available datasets, including RNA-seq from E7.5 reciprocal hybrid C57BL6/CAST Epi and ExE, and H3K4me3, H3K36me3, and H3K27me3 ChIP-seq in E6.5 C57BL6/CAST ExE[18], were used in this study (available in Gene Expression Omnibus series GSE124216). Bisulphite-sequencing datasets for E3.5 inner cell mass and trophectoderm were also used in this study[16] (available in series GSE84236). All raw data files were mapped and trimmed using the pipelines described below. Gene expression patterns of *Sox2* in pre-implantation embryos were evaluated using publicly available database https://endoderm-explorer.com/[47].

## Data mapping and QC

Fastq sequence files were quality and adaptor trimmed with trim galore v0.4.2 using default parameters. For PBAT libraries, the -clip option was used to remove the bases covered by the PBAT primer. Mapping of ChIP-seq data was performed with Bowtie v2.2.9 against the *Mus musculus* (GRCm38) derived genome, where SNPs between hybrid strains (C57BL6 and CAST/Ei) had been masked by the ambiguity nucleobase N (N-masked genome). The resulting hits were filtered to remove mappings with a MAPQ scores < 20. RNAseq data was subjected to trimming with Trim Galore (v0.6.6, options '--clip_r1 8' to remove a nucleotide bias at the 5' end), and aligned to the GRCm38 mouse genome build, to which a ribosomal DNA sequence had been added (BK000964.1), using HISAT2 (v2.1.0, guided by gene models from Ensembl annotation release 90, options: --dta –no-softclip). Hits were again filtered to remove mappings with MAPQ scores <20. Sequencing depths for all libraries are provided in Supplementary Data 1.

Mapping and methylation calling of bisulphite-seq data was performed using Bismark v0.16.3 in PBAT mode against the mouse GRCm38 genome assembly. Trimmed reads were first aligned to the genome in paired-end mode to be able to detect and discard overlapping parts of the reads while writing out unmapped singleton reads; in a second step remaining singleton reads were aligned in single-end mode. Alignments were carried out with Bismark[48] with the following set of parameters: a) paired-end mode: --pbat; b) single-end mode for Read 1: --pbat; c) single-end mode for Read 2: defaults. Reads were then deduplicated with deduplicate_bismark selecting a random alignment for positions that were covered more than once. Following methylation extraction, CpG context files from PE and SE runs were used to generate a single coverage file (the "Dirty Harry" procedure). Based on DNA methylation levels at CHH sequences, bisulphite conversion efficiencies were estimated to be average of 96.5% (+/−1.9%).

Mapping and quantitation of 10X single nuclei RNA-Seq data was performed with CellRanger v6.0.0 against the gex-mm10-2020-A transcriptome using the --include-introns option.

## Bisulphite-seq analysis

DNA methylation data was evaluated using 100-CpG windows, excluding the mitochondria (MT), X and Y chromosomes. Percentage DNA methylation was calculated per base and then averaged per window in each replicate with a minimum of 10 informative CpGs required to quantitate each window. Methylation values were then averaged between replicates for each group. Differentially methylated regions (DMRs) were identified using two methods in SeqMonk v1.48.0, logistic regression with a minimum 20% difference in DNA methylation between KO and WT was used to identify DMRs in each KO. To identify genomic regions most affected in each KO model, a binomial test was used to identify the subset of domains that showed the largest differences in DNA methylation in each of the *Dnmt3a*, *Dnmt3b* and *Dnmt3l* KOs; these DMRs were used for the UMAP analysis, described below.

## UMAP dimensionality reduction and clustering

Epigenetic data, including DNA methylation, H3K4me3, H3K27ac, H3K4me1, H3K36me3, and H3K27me3, for autosomal 100-CpG windows were used in UMAP dimensionality reduction and clustering. Regions were quantitated as RPKM for WT samples with coverage outliers excluded. To reduce the impact of outliers and correct for variable ChIP enrichment, capped, scaled and centred values were used for randomly initialised UMAP (n_neighbors=10, min_dist=0.05)[49]. The same input was used for Expectation Maximisation Clustering in Weka[50], setting the number of clusters to 15. The resulting clusters were summarised into the functional annotations shown in Fig. 2 based on similarity of histone modifications. Enrichment or depletion of the most hypomethylated DMRs at chromatin features in *Dnmt3a*, *Dnmt3b* and *Dnmt3l* KOs were statistically compared to a random set of 100-CpG windows using one-proportion Z test with Yates' continuity correction (results presented in Supplementary Data 2).

## RNA-seq analysis

Gene expression was quantitated using the RNA-seq quantitation pipeline in SeqMonk. Differentially expressed genes were identified using DESeq2 in SeqMonk, with a minimum 1.5-fold-change between KO and matched WT samples. Gene ontology analyses were done using DAVID Bioinformatics Resource (https://david.ncifcrf.gov/), using the default settings and the addition of the UP-TISSUE category. The GRCm38 repeatmasker annotations were used to evaluate expression of all ERV repeat classes. Informative ERV LTRs were defined as those with one same-stranded read in at least one RNA-seq replicate (N = 8142). ERV expression was quantitated as RPKM using only reads that mapped to the same strand as the ERV. LIMMA statistic (p < 0.05 correcting for multiple comparisons) in SeqMonk v1.48.0 was used to identify ERVs that showed significantly different expression between *Dnmt3a/b* DKO and WT controls.

## Single-nuclei RNA-seq analysis

Filtered count matrices from CellRanger were imported and analysed using Seurat v3. Cells containing more than 0.5% mitochondrial reads were removed, then standard CLR normalisation, VST variable gene selection (1000 genes), scaling, PCA and tSNE (15 PCs, perplexity = 100). Cell clusters were defined using Louvain clustering (resolution 0.3), and clusters deriving from maternal cells were identified based on their *Xist* expression, and discarded.

Matches to known placental cell types were predicted using SCINA (sensitivity_cutoff=0.9) with the 50 most significant markers per cell type from a previous study[25].

## Statistical analysis

Statistical tests were performed in Excel, R v4.2.0, http://vassarstats.net/, and GraphPad Prism 9. For linear correlations, Pearson Correlation Coefficients are provided as R-values and linear correlation statistical test was used to calculate p-values. Pairwise comparisons of *Dnmt* gene expression between E7.5 Epi and ExE, MCT4 and MCT1 fluorescence intensity, foetal weights, placental and labyrinth size and proportions between KOs and WT controls were done using a standard two-tailed t-test statistic, correcting the significance threshold using Bonferroni correction for multiple comparisons within each analysis. Comparisons of frequency data were done using Chi-square statistic (Supplementary Fig. 3g) and Fisher's Exact test (Fig. 4f). ERV lengths across subclasses were compared using a non-parametric Mann Whitney test (Supplementary Fig. 3h). Comparison of fold change expression of *Dnmt3a/b* up-regulated ERVs (N = 23) between *Dnmt* KOs was done using an ANOVA statistic, with *post hoc* Tukey Test to determine significant pairwise comparisons. Figures were generated using Excel, SeqMonk v1.48.0, R Studio v1.1.456, R v4.2.0 (R package ggplot2), GraphPad Prism 9, CellRanger v6.0.0, and https://endoderm-explorer.com/.

## Reporting summary

Further information on research design is available in the Nature Portfolio Reporting Summary linked to this article.

## Data availability

The data that support this study are available from the corresponding author upon reasonable request. The sequencing data generated in this study has been deposited in Gene Expression Omnibus (GEO) database under accession number GSE203462, including raw and processed data files. Publicly available RNA-seq and ChIP-seq datasets (GSE124216) and bisulphite-seq data (GSE84236) were used in this study. Publicly available single-cell RNA-seq data was accessed through https://endoderm-explorer.com/. Source data are provided with this paper.

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

## Acknowledgements

Thank you to the Biological Support Unit, Flow Cytometry Facility, Dr. Paula Kokko-Gonzales and Dr. Amelia Edwards at the Sequencing Facility, Dr. Simon Walker at the Imaging Facility, and Dr. Felix Krueger and Dr. Laura Biggins in the Bioinformatics Facility at the Babraham Institute. Thank you to Dr. Tim Lohoff and Dr. Stephen Clark for advice in establishing the single-nuclei isolation protocol. Thank you to Prof. Kathy Niakan, Director of the Centre for Trophoblast Research, and Dr. Georgia Lea, postdoctoral researcher in the Department of Physiology, Development and Neuroscience, at the University of Cambridge, for their valuable comments on the manuscript. This work was funded by a Next Generation Fellowship from the Centre for Trophoblast Research, Career Support Fund from the University of Cambridge, and a Sir Henry Dale Fellowship from the Royal Society and Wellcome Trust (222582/Z/21/Z), awarded to CH. VPG is supported by the Ministerio de Ciencia e Innovación (RYC-2019-026956 and PID2020-114459RA-I00). Part of the equipment employed in this work was funded by Generalitat Valenciana and co-financed with ERDF funds (OP ERDF of Comunitat Valenciana 2014–2020). MH is supported by a Tier I Canada Research Chair in Developmental Genetics and Epigenetics.

## Author contributions

C.H. conceptualised the project with valuable input from M.H. and W.D. C.H., M.M.L. and V.P.G. collected data and performed experiments. S.A., C.K., V.P.G., and C.H. performed data analysis and generated figures. S.A., C.K., W.D., M.H., V.P.G., and C.H. contributed to data interpretation. C.H. wrote the manuscript with contributions from S.A., C.K., W.D., M.H. and V.P.G.

## Competing interests

The authors declare no competing interests.
