## [Peer Review File · Nature Communications]

Mechanisms and function of de novo DNA methylation in placental development reveals an essential role for DNMT3BREVIEWER COMMENTS

Reviewer #1 (Remarks to the Author):

Two partially redundant de novo DNA methyltransferases, DNMT3A and 3B, carry out the embryonic DNA methylation program that occurs during epigenetic reprogramming in early mammalian embryogenesis. While mice harboring mutations in both genes exhibit embryonic lethality post-gastrulation, single mutations lead to disparate levels of severity, with *Dnmt3B*^{-/-} mice dying in utero, and *Dnmt3A*^{-/-} mice dying post-natally. Finally, the non-catalytic paralog, DNMT3L, is absolutely required for germline DNA methylation, but appears dispensable in the embryo, and homozygous *Dnmt3L*^{-/-} mice do not exhibit obvious defects apart from sterility. The precise reason for the different developmental phenotypes remains a highly sought-after question in the mammalian DNA methylation field. In this study Andrews et al., describe the methylation and expression in the epiblast and extra embryonic tissues (Exe) of the *Dnmt3a*, *Dnmt3b*, *Dnmt3l* and *Dnmt3s/b* KO mice. They show that in the *Dnmtb* KO the mouse placenta fails to develop properly. Using a *Sox2*-Cre conditional KO the authors recover the placenta phenotype and the embryonic lethality, thus demonstrating that de novo methylation established by DNMT3B is crucial in the trophoblast for proper placental development and embryo survival.

To our mind, there are two substantial findings from this study: 1. *Dnmt3L* mutant placenta exhibit substantial DNA hypomethylation, as opposed to the embryo proper, where the methylome appears largely unperturbed, and 2. DNMT3B is necessary for proper placental development and that it is the main cause of embryonic lethality in the *Dnmtb* KO embryos. The first point was unfortunately not discussed in detail (see major point), but raises the interesting question of why DNMT3L would display different activities, given its expression in the embryo. The second point came as quite a surprise (at least to us), as we had assumed the severe *Dnmt3B* phenotype was due to misregulation in the embryo. These data will be of interest for the Epigenetics and placenta biology fields. However, there are some points that the authors should address before publication of this manuscript.

Major points:

- The authors should show methylation data of the *Dnmt3b* KO placentas E12.5. Decato et al., Mol Bio 2017 showed that the placenta has an important within-tissue methylation variance that is higher than in any other differentiated tissue. We wonder if the methylation defects that they observed in the E7.5 embryos are more aggravated in the labyrinth cells than in other cells of the placenta thus causing the phenotype that the authors observe; this data could also complement the single cell expression data as maybe part of the phenotype could be explained because of the misregulation of genes that cannot be detected using this technique. As it has been reported before that misregulation of epigenetic marks in placenta is correlated pregnancy complications, the DNA methylation data that the authors can generate with the material they have could give insight on how the epigenetic landscape affects placenta and embryo development. Furthermore, by doing this experiment the authors will corroborate the importance of DNA methylation in keeping germline genes silenced in placenta.

- I found there to be an overall lack of discussion of the *Dnmt3l* mutant placenta phenotype. Could the authors speculate about why there is a greater loss of DNA methylation in the placenta than the embryo in these mutants. One hypothesis that came to our mind was presence or absence of DNMT3B isoforms. It has been demonstrated that at least one DNMT3B isoform is catalytically inactive and can behave like DNMT3L (PMID: 33004415). Could the authors mine their (or other) RNAseq data to see if this isoform is present in the embryo proper but not the placenta? Or perhaps offer other insights? And secondly, given the DNA hypomethylation phenotype, could the authors explain the lack of a placental phenotype in the *Dnmtl* KO seeing as *Dnmt3B* mutants do have such a phenotype and the authors established a link between DNMT3B and DNMT3L in intergenic regions.

Minor points:

Figure 1: there is no figure 1d in the panel but it is in the legend.

Figure 2a. It would be interesting to add the DNA methylation tracks of the different KOs; this would help to understand visualize the loss of methylation in the context of the chromatin landscape.

Figure 2e: The authors chose the UMAP clustering figures to show where the hypomethylated DMRs of the different KO were in the different chromatin features described in the figure 2b. However, the visualization of the data is a bit difficult using this kind of graphs, which readers are more accustomed to seeing represent single cell data. I recommend the authors to show the changes in methylation in a more intuitive way like the one used in the figure 2d to facilitate reader comprehension.

Lines 126-127: The authors suggest that DNMT3L modulates the localization of DNMT3B in unmarked intergenic regions; it would be interesting to see if these regions that are unmethylated in the Dnmt1 KO are also unmethylated in the Dnmt3b KO to demonstrate a real correlation as in the figure 2e it is not clear enough.

Figure 3a. As the transcriptomes of the different KO are similar, they don't cluster at all in the PCA analysis. As they want to show this in a main figure, I recommend the authors showing this lack of changes between samples using another kind of graph like a correlogram instead of the PCA graph, as I consider it would be more informative.

Lines 149-158. Endogenous retroviruses have been reported to act as enhancers in mouse extraembryonic tissue. The authors identified 23 ERV that are upregulated in their RNA-seq however they do not comment on the methylation state of these genes in the DKO; it would be interesting to see if the upregulation of these ERV is correlated with the level of methylation (something similar of what they show in the figure 3d.). In addition, the authors did not describe the transcription levels of this ERV in the single mutants. Are any of these ERVs misregulated in the Dnmt3B KO?

Lines 171-172. Extended data 4c. In this figure the authors show the area of the placentas and the labyrinth in the WT and the Dnmt3b KO placentas. We think they should add to these figures the information about the Dnmt3l and Dnmt3a placentas as it is more quantitative and would complement the staining of the extended data Fig4a. The placental labyrinth is the area where the gas and nutrient exchange between the maternal and the fetal blood occurs. We wonder if the authors did notice an impaired growth of the Dnmt3b KO embryos at this stage and if they can comment on this in the manuscript. It is likely that this is happening due to the fact of having less exchange surface in the underdeveloped labyrinth zone.

General comments:

In some of the extended data figures the title of the axes are not legible.

Reviewer #2 (Remarks to the Author):

This study by Hanna and colleagues reports on the role of de novo DNA methylation in placental lineage specification and development. The first part of study focuses on how loss of Dnmt3a/b/l affects gene expression and methylation landscape in early development. While they identify epigenetic changes, the effects on gene expression is not very robust (only 17 germ line genes are de-repressed). These findings appear not to have significant biological implications on placenta lineage specification as the study nicely describes. However the most novel and super exciting part of the

study is the second part, which makes it of high impact to the field. They show that the biologically critical function of Dnmt3b during embryogenesis is essentially in the placenta as deletion of Dnmt3b in ICM/embryo proper only (but not in the trophoblast/placenta) supports embryonic development till late gestation/birth. They show that Dnmt3b is important, not for specification of trophoblast and placental cell types, but rather for the placental layer formation and cellular function. The findings in the second part are very interesting and hence the study should be considered for publication. However the following points need to be addressed:

1. Must-do experiment to support the conclusions: In the Sox2-Cre model embryos need to be genotyped properly, to make sure DNMT3B is completely deleted in the whole embryo proper (genotyping only the tail tip of the embryo is not enough, DNA from the whole embryo should be genotyped by PCR, RTqPCR or Western to make sure there is no residual Dnmt3b in any embryo proper tissues (and that other Dnmts are not induced). The same holds true for genotyping the placenta. It should be shown that in Sox2-Cre placentas Dnmt3b is expressed at normal levels.
2. Must-do experiment to support the conclusions: A better comparison of Sox2-Cre Dnmt3f/f embryos with Dnmt3b KO embryos is needed. Only some pictures/weights are shown. It is not quite clear if the E18.5 embryos (which are smaller in size) are born alive and how long they survive. That data is not clearly presented (only mentioned in the text). Clearly deletion of Dnmt3b in embryo proper alone has some effects on growth (which is independent of placenta). The E18.5 embryos in the picture seem very unhealthy and would be of interest to know what happens to them at birth and how long they live (in what condition).
3. Would-have-been-nice study to do to support the conclusion (not necessary if there are technical challenges): If Dnmt3b was deleted in the trophoblast only (for example by viral infection of Cre in Dnmt3b^{fl/fl} blastocyst trophoblast followed by implantation), those embryos would have died mid-gestation with placental defects. This would have clearly supported the conclusions.
4. Must comment on in the discussion section: An important study also published in Nature Communications in 2019 [Nowailis et al "Catalytically inactive Dnmt3b rescues mouse embryonic development by accessory and repressive functions"] implicates non-catalytic functions of Dnmt3b in viability of mice, it may have some mechanistic implications to this study and would be good to be discussed in detail in the context of how deregulation of placental genes in the placenta are regulated by DNA methylation vs other repressive functions of Dnmt3b.
5. Must comment on in the discussion section as future studies: Deletion with Sox2-Cre in 3a/b double conditional embryos would have been very useful. 3a/b DKO die early before placenta formation but it is likely that is also due to placental defects (and Sox2-Cre model would have answered it).

The study provides mechanism but the mechanism is somewhat less defined and blurry. The first part doesn't connect with the second part as the authors nicely acknowledge. Lineage specification is not affected but placental cell type function and architecture are affected. This is not due to early embryonic events, rather later events in the placenta. Not sure if de-repression of germ cell genes causes the defects in placenta or deregulation of placental genes. Likely both but the how part of it is unknown. There is no elucidation on what type of transcription programs/genes are disrupted due to deregulation of placental genes and how that affects placental layers formation. Some comments on the roles of these deregulated genes in placental layer formation will be useful.

Reviewer #3 (Remarks to the Author):

Review: NCOMMS-22-22567-T

Title: Mechanisms and function of de novo DNA methylation in placental development

Comments to the authors:

The present study by Andrews et al investigated the impact of de novo DNMTs on the placenta trophoblast cell lineage during embryogenesis. This study makes important contributions to the field of placental epigenetics. In particular, the authors provide evidence to support that Dnmts 3A/B/L are essential for the establishment of the placental methylome, and that they exert their functions, in part, by targeting specific chromatin states. This highlights the interplay between different types of epigenetic modifications that are oftentimes studied in isolation. Additionally, this study demonstrates that a loss of Dnmt3B in particular is associated with lethality due to impaired placental labyrinth formation, with data suggesting this is due to transcriptional changes in genes typically expressed during gametogenesis, rather than issues with cell lineage specification or differentiation. This manuscript was well written, and easy to follow, and is appropriate for the audience of Nature Communications. The figures, extended figures, and supplemental tables were easy to interpret. Particular strengths of this study include the assessment of the interactions between DNA methylation and histone modifications, and the delineation of the important role of DNMTs, particularly DNMT3B in placental formation and function. This work is significant given the important public health concerns associated with placental dysfunction during pregnancy that the authors highlight. However, there are some suggested changes and clarifications that I would like the authors to address to strengthen their manuscript.

Major Comments:

1) While the methods section is very detailed, there is no statistical analysis section. Additionally, there are few statistical tests that were performed/reported in the presentation of the data. While the figures are clear and well-presented, the limited analysis makes it difficult to interpret the findings and draw accurate conclusions at times. Please address the following points about statistical analyses/presentation of data below:

- In the bean plots in Figure 2c the legend describes "enrichment" though no statistical test was performed to support this analysis and interpretation.
- Bean plots in Figure 2d analyzing methylation, no analysis was performed to help interpret/bolster the meaning of these findings.
- There is an R value reported in Figure 3b, but there is no description of what type of regression was run to achieve this value. Also, is this an r or R²?
- Figure 3d, an r value is reported but no description of the regression that was run to derive the r value, and no p-value for the correlation is provided in figure legends. In the text (Line 148), a p-value is reported, but we don't know what type of statistic was run.
- Extended data Figure 3a-c, please include type of regression run to generate R value. There are no error bars for extended data Figure 3g.
- Figure 4b, can you quantify the difference in IF intensity to provide a quantitative difference in MCT1/4 staining between WT and KO, and then run a statistical analysis on this difference? This would bolster the compelling finding that was presented in this Figure.
- Extended data Figure 5f – is this a significant finding/pattern of higher expression in the KO or just an observation?
- Figure 5c, there are no error bars on the box plot even though these are frequencies calculated across litters.

2) The current study is focused on the role of de novo DNMTs 3A and B, and the requisite cofactor 3L. However, there is no mention of the maintenance enzyme DNMT1. This seems important given that cell division and proliferation are occurring as the placenta develops, and thus, any existing methylation marks would be maintained across cell divisions by DNMT1. Further, compensatory de novo methylation by DNMT1 has been reported in certain situations, particularly repetitive elements, which were investigated in the present manuscript (see PMID 34140676 for reference). Given that DNMT1 is expressed in the placenta, please address why a DNMT1 KO was not included in this study and how this might limit or contribute to the interpretation of the findings presented in this

manuscript.

Minor Comments:

Introduction

Line 35: "DNA methylation is a repressive epigenetic...". While this is largely true, it is an over characterization/simplification of the role of DNA methylation in regulating transcription as there are times where more methylation is associated with increased gene expression. Please modify this language.

Line 87: Can authors explain why they did not include a 3A/3L or 3B/3L double knockout in their studies?

Results

Lines 68-96: There is no mention of Figure 1d in the text.

Line 85: Please explain why 20% methylation difference was chosen.

Lines 97-100 & 123-127: These types of summaries are more appropriate for the discussion than the results section.

Line 148/Figure 3d: The x-axis is a little confusing for this figure. Consider plotting the WT and KO regressions separately, but overlapping on the same figure, to show the difference in the direction of the association based on Dnmt status.

Lines 176-179: Sentence beginning "branching morphogenesis..." is a little unclear grammatically. Please edit.

Discussion

Line 244: Specify this as a primary role of DNA methylation in the placenta.

Can the authors expand upon why they observed more severe phenotypes with the Dnmt3b KO than the Dnmt3a knockout?

Methods

See above about including statistical analysis section to the methods.

POINT-BY-POINT RESPONSES TO REVIEWER COMMENTS:

Reviewer #1 (Remarks to the Author):

Two partially redundant de novo DNA methyltransferases, DNMT3A and 3B, carry out the embryonic DNA methylation program that occurs during epigenetic reprogramming in early mammalian embryogenesis. While mice harboring mutations in both genes exhibit embryonic lethality post-gastrulation, single mutations lead to disparate levels of severity, with *Dnmt3B*^{-/-} mice dying in utero, and *Dnmt3A*^{-/-} mice dying post-natally. Finally, the non-catalytic paralog, DNMT3L, is absolutely required for germline DNA methylation, but appears dispensable in the embryo, and homozygous *Dnmt3L*^{-/-} mice do not exhibit obvious defects apart from sterility. The precise reason for the different developmental phenotypes remains a highly sought-after question in the mammalian DNA methylation field. In this study Andrews et al., describe the methylation and expression in the epiblast and extra embryonic tissues (Exe) of the *Dnmt3a*, *Dnmt3b*, *Dnmt3l* and *Dnmt3s/b* KO mice. They show that in the *Dnmtb* KO the mouse placenta fails to develop properly. Using a *Sox2*-Cre conditional KO the authors recover the placenta phenotype and the embryonic lethality, thus demonstrating that de novo methylation established by DNMT3B is crucial in the trophoblast for proper placental development and embryo survival.

To our mind, there are two substantial findings from this study: 1. *Dnmt3L* mutant placenta exhibit substantial DNA hypomethylation, as opposed to the embryo proper, where the methylome appears largely unperturbed, and 2. DNMT3B is necessary for proper placental development and that it is the main cause of embryonic lethality in the *Dnmtb* KO embryos. The first point was unfortunately not discussed in detail (see major point), but raises the interesting question of why DNMT3L would display different activities, given its expression in the embryo. The second point came as quite a surprise (at least to us), as we had assumed the severe *Dnmt3B* phenotype was due to misregulation in the embryo. These data will be of interest for the Epigenetics and placenta biology fields. However, there are some points that the authors should to address before publication of this manuscript.

Major points:

- The authors should show methylation data of the *Dnmt3b* KO placentas E12.5. Decato et al., Mol Bio 2017 showed that the placenta has an important within-tissue methylation variance that is higher than in any other differentiated tissue. We wonder if the methylation defects that they observed in the E7.5 embryos are more aggravated in the labyrinth cells than in other cells of the placenta thus causing the phenotype that the authors observe; this data could also complement the single cell expression data as maybe part of the phenotype could be explained because of the misregulation of genes that cannot be detected using this technique. As it has been reported before that misregulation of epigenetic marks in placenta is correlated pregnancy complications, the DNA methylation data that the authors can generate with the material they have could give insight on how the epigenetic landscape affects placenta and embryo development. Furthermore, by doing this experiment the authors will corroborate the importance of DNA methylation in keeping germline genes silenced in placenta.

Thank you to the reviewer for this suggestion - it may be that DNA methylation loss is exacerbated in some nuclei, which could underpin some of the heterogeneity we see in the loss of MCT4 across the *Dnmt3b* KO labyrinth. To address this question, a single-nuclei bisulphite-sequencing (as opposed to single-cell bisulphite-seq) approach is necessary. This would be necessary due to the syncytial nature of the relevant trophoblast population in the labyrinth but

unfortunately, this is a method that has not yet been developed by the field. This presents an exciting potential future experiment, once the technique is available.

To determine whether there is evidence for a difference in de-repression of methylation-sensitive germline genes between placental compartments, we further analysed our single-nuclei RNA-seq data (added lines 197-200, Fig. 4f). We find that upregulation of these genes is similarly significant in both the junctional zone and labyrinth (Fig. 4f). These findings may suggest that the region-specific perturbations in developmental processes are underpinned by the sensitivity of these processes to the presence of germline factors, rather than cell type-specific de-repression.

- I found there to be an overall lack of discussion of the Dnmt3l mutant placenta phenotype. Could the authors speculate about why there is a greater loss of DNA methylation in the placenta than the embryo in these mutants. One hypothesis that came to our mind was presence or absence of DNMT3B isoforms. It has been demonstrated that at least one DNMT3B isoform is catalytically inactive and can behave like DNMT3L (PMID: 33004415). Could the authors mine their (or other) RNAseq data to see if this isoform is present in the embryo proper but not the placenta? Or perhaps offer other insights? And secondly, given the DNA hypomethylation phenotype, could the authors explain the lack of a placental phenotype in the Dnmtl KO seeing as Dnmt3B mutants do have such a phenotype and the authors established a link between DNMT3B and DNMT3L in intergenic regions.

We agree, the differing requirement for DNMT3L in the epiblast and ExE is intriguing. As suggested, we evaluated whether there was differing expression of *Dnmt3b* isoforms between E7.5 epiblast and ExE, but found no evidence for this (Reviewer Figure 1 below).

Figure 1. Barplot shows the relative proportions of *Dnmt3b* isoforms of total *Dnmt3b* transcripts in E7.5 epiblast (N=2) and ExE (N=2), quantified using default settings in SeqMonk version 1.48.0 and merging isoforms with the same starting exon. Error bars show standard deviation between datasets.

We speculate that the lower overall expression of *de novo* DNMT3A and 3B in trophoblast (Extended data Fig. 1a) may necessitate DNMT3L to boost their catalytic activity and DNA binding (Gower *et al.* 2005 *JCB*; 280:13341-8). We have added this to the discussion (lines 265-266).

We have added a direct comparison of *Dnmt3l* and *Dnmt3b* KO DNA methylation, which indeed shows they have remarkably similar genome-wide

patterns (added Extended Data Fig. 2d). The notable exceptions are the germline gene promoters (Fig. 3c), which lose less DNA methylation in the *Dnmt3l* KO than the *Dnmt3b* KO and consequently fail to become significantly de-repressed in the *Dnmt3l* KOs (Extended Data Fig. 3c). These data support that these loci likely underpin the phenotype we are observing in the *Dnmt3b* KO.

Minor points:

Figure 1: there is no figure 1d in the panel but it is in the legend.

This error has been corrected (line 676).

Figure 2a. It would be interesting to add the DNA methylation tracks of the different KOs; this would help to understand visualize the loss of methylation in the context of the chromatin landscape.

Annotation tracks for the *Dnmt* DMRs have been added to Fig. 2a.

Figure 2e: The authors chose the UMAP clustering figures to show where the hypomethylated DMRs of the different KO were in the different chromatin features described in the figure 2b. However, the visualization of the data is a bit difficult using this kind of graphs, which readers are more accustomed to seeing represent single cell data. I recommend the authors to show the changes in methylation in a more intuitive way like the one used in the figure 2d to facilitate reader comprehension.

To improve the clarity of the analysis presented in Fig. 2, we have exchanged Fig. 2d with Extended data Fig. 2c. The bar plot (now Fig. 2d) is a more typical way to present quantification of relative enrichment of chromatin features for each of the DMRs and should hopefully be more intuitive. The statistical analysis for this bar plot has also been added as Supplementary Table 2.

Lines 126-127: The authors suggest that DNMT3L modulates the localization of DNMT3B in unmarked intergenic regions; it would be interesting to see if these regions that are unmethylated in the *Dnmtl* KO are also unmethylated in the *Dnmt3b* KO to demonstrate a real correlation as in the figure 2e it is not clear enough.

We have added a figure correlating DNA methylation of 100-CpG windows between the *Dnmt3b* and *Dnmt3l* KOs (Extended data Fig. 2d) to better enable direct comparison of these two KOs. This analysis is described in the results (lines 119-122).

Figure 3a. As the transcriptomes of the different KO are similar, they don't cluster at all in the PCA analysis. As they want to show this in a main figure, I recommend the authors showing this lack of changes between samples using another kind of graph like a correlogram instead of the PCA graph, as I consider it would be more informative.

We have included correlations between each of the KOs with their respective WT controls in Extended data Fig. 2 (R-values>0.99). We find that the lack of clustering on the PCA plot of the RNAseq replicates can be directly contrasted to the PCA plot we show for the DNA methylation data, which shows distinct clustering of each KO (Fig. 1b), hence we have kept both representations of these data.

Lines 149-158. Endogenous retroviruses have been reported to act as enhancers in mouse extraembryonic tissue. The authors identified 23 ERV that are upregulated in their RNA-seq however they do not comment on the methylation state of these genes in the DKO; it would be interesting to see if the upregulation of these ERV is correlated with the level of methylation (something similar of what

they show in the figure 3d.). In addition, the authors did not describe the transcription levels of this ERV in the single mutants. Are any of these ERVs misregulated in the Dnmt3B KO?

As suggested, we have added the comparison between loss of DNA methylation at ERVs with up-regulation of expression across all of the KOs (Extended data Fig. 3i, results section lines 156-161). Interestingly, de-repression of ERVs appears to be much more bimodal, rather than linear as is seen for the germline genes. This suggests that for ERVs, a substantial loss of DNA methylation is required for upregulation, whereas germline gene promoters appear to be more sensitive to incremental DNA methylation losses.

Lines 171-172. Extended data 4c. In this figure the authors show the area of the placentas and the labyrinth in the WT and the Dnmt3b KO placentas. We think they should add to these figures the information about the Dnmt3l and Dnmt3a placentas as it is more quantitative and would complement the staining of the extended data Fig4a. The placental labyrinth is the area where the gas and nutrient exchange between the maternal and the fetal blood occurs. We wonder if the authors did notice an impaired growth of the Dnmt3b KO embryos at this stage and if they can comment on this in the manuscript. It is likely that this is happening due to the fact of having less exchange surface in the underdeveloped labyrinth zone.

Thank you for the reviewer's suggestions, we have added quantitative comparisons for *Dnmt3l* and *Dnmt3a* of placental and labyrinth size (Extended data Fig. 4a and b). We have also added an image of an example *Dnmt3b* KO E12.5 embryo to Fig. 5d to show the extent of developmental delay at this time point, in direct contrast to the *Dnmt3b* cKO and WT.

General comments:

In some of the extended data figures the title of the axes are not legible.

We have increased the font sizes for all illegible axes.

Reviewer #2 (Remarks to the Author):

This study by Hanna and colleagues reports on the role of de novo DNA methylation in placental lineage specification and development. The first part of study focuses on how loss of Dnmt3a/b/l affects gene expression and methylation landscape in early development. While they identify epigenetic changes, the effects on gene expression is not very robust (only 17 germ line genes are de-repressed). These findings appear not to have significant biological implications on placenta lineage specification as the study nicely describes. However the most novel and super exciting part of the study is the second part, which makes it of high impact to the field. They show that the biologically critical function of Dnmt3b during embryogenesis is essentially in the placenta as deletion of Dnmt3b in ICM/embryo proper only (but not in the trophoblast/placenta) supports embryonic development till late gestation/birth. They show that Dnmt3b is important, not for specification of trophoblast and placental cell types, but rather for the placental layer formation and cellular function. The findings in the second part are very interesting and hence the study should be considered for publication. However the following points needs to be addressed:

1. Must-do experiment to support the conclusions: In the Sox2-Cre model embryos need to be genotyped properly, to make sure DNMT3B is completely deleted in the whole embryo proper (genotyping only the tail tip of the embryo is not enough, DNA from the whole embryo should be genotyped by PCR, RTqPCR or Western to make sure there is no residual Dnmt3b in any embryo proper tissues (and that

other Dnmts are not induced). The same holds true for genotyping the placenta. It should be shown that in Sox2-Cre placentas Dnmt3b is expressed at normal levels.

We have performed several additional experiments to confirm the specificity the Sox2-Cre model:

- (1) We have now genotyped whole embryos and placentas from Sox2-Cre *Dnmt3b* cKOs at E8.5 by PCR to confirm that *Dnmt3b* is absent from all embryonic cells, while the floxed allele remains intact in placenta. This data has been added to Extended data Fig. 6d.
- (2) We performed an additional 10X single-nuclei RNA-sequencing experiment on a Sox2-Cre *Dnmt3b* cKO and matched WT control E12.5 placentas to demonstrate that there is no difference in gene expression, including methylation-sensitive germline genes, in the cKO placenta. This data has been added as Fig. 5c and is described in the results (lines 225-227).
- (3) Finally, we quantified MCT1 and MCT4 staining in *Dnmt3b* cKO E12.5 placentas compared to *Dnmt3b* KO and WT littermates to demonstrate that the 'rescue' of the placental phenotype is indeed complete. This data has been added as Extended data Fig. 6e.

We find that our data, in support of the complete penetrance and specificity of lacZ staining in epiblast reported by the McMahon lab (Hayashi *et al.* 2002 *Mech Dev*; 119:S97-S101), provide compelling evidence that the Sox2-Cre is highly specific and penetrant.

2. Must-do experiment to support the conclusions: A better comparison of Sox2-Cre *Dnmt3f/f* embryos with *Dnmt3b* KO embryos is needed. Only some pictures/weights are shown. It is not quite clear if the E18.5 embryos (which are smaller in size) are born alive and how long they survive. That data is not clearly presented (only mentioned in the text). Clearly deletion of *Dnmt3b* in embryo proper alone has some effects on growth (which is independent of placenta). The E18.5 embryos in the picture seems very unhealthy and would be of interest to know what happens to them at birth and how long they live (in what condition).

We agree that more comparative data will improve the interpretation of the *Dnmt3b* cKO. Therefore, we have added representative images of *Dnmt3b* KO and cKO embryos and WT littermate controls at E12.5, demonstrating the extent of 'rescue' of the embryonic phenotype at mid-gestation through the presence of a functional placenta in the *Dnmt3b* cKO (Fig. 5d). We have quantitatively compared the MCT1/4 staining in the *Dnmt3b* KO and cKO E12.5 placentas, highlighting that placental morphology appears completely restored in the *Dnmt3b* cKO (Extended data Fig. 6e). We have also added embryonic weights for the *Dnmt3b* cKO embryos at E12.5, showing that the embryonic phenotype in the *Dnmt3b* cKO only becomes evident in late gestation (Fig. 5f). Finally, we have added the frequency data for the *Dnmt3b* cKO from 6 litters at postnatal day 10, showing that these embryos die perinatally (Fig. 5e). Future studies will be valuable in assessing the abnormal morphology of *Dnmt3b* cKO embryos before birth to determine the cause of perinatal lethality, but to do this comprehensively is beyond the scope of this study.

3. Would-have-been-nice study to do to support the conclusion (not necessary if there are technical challenges): If *Dnmt3b* was deleted in the trophoctoderm only (for example by viral infection of Cre in *Dnmt3bf/f* blastocyst trophoctoderm followed by implantation), those embryos would have died mid-gestation with placental defects. This would have clearly supported the conclusions.

We agree that this would have been a nice experiment. A pan-trophoblast Cre remains a challenge in the field, as those that use endogenous promoters (e.g., *Cdx2*, *Cdx1*, *Elf5*) are expressed in other tissues during development. We imported a strain that looked promising, developed by Gustavo Leone's lab that utilises a human promoter for CYP19 (Wenzel and Leone. 2007 *Genesis*; 45:129-34). Using a tdTomato/EGFP reporter line, we find that CYP19-Cre is robustly expressed in ExE at E6.5, but at E5.5 we find no reporter expression (Reviewers Figure 2 below).

Figure 2. Immunofluorescence imaging of E5.5 (left) and E6.5 (right) live embryos carrying CYP19-Cre, a pan-trophoblast-specific Cre (Wenzel and Leone. 2007) and the ROSA^{mT/mG} reporter. In cells not expressing Cre recombinase, tdTomato fluorescence is expressed and localised to the cell membrane (red). Cre recombinase expressing cells (and future cell lineages derived from these cells) excise the tdTomato gene, and thus express EGFP that is also localised to the cell membrane (green).

Thus, unfortunately, a trophoblast-specific KO for *Dnmt3b* using CYP19-Cre would not delete the locus before the majority of *de novo* DNA methylation has already taken place. We hope to be able to do this experiment in future, when the genetic tools are available.

4. Must comment on in the discussion section: An important study also published in Nature Communications in 2019 [Nowailis et al “Catalytically inactive Dnmt3b rescues mouse embryonic development by accessory and repressive functions”] implicates non-catalytic functions of Dnmt3b in viability of mice, it may have some mechanistic implications to this study and would be good to be discussed in detail in the context of how deregulation of placental genes in the placenta are regulated by DNA methylation vs other repressive functions of Dnmt3b.

Thank you, this is an important study to highlight in our discussion. We have discussed their findings in lines 274-280.

5. Must comment on in the discussion section as future studies: Deletion with Sox2-Cre in 3a/b double conditional embryos would have been very useful. 3a/b DKO die early before placenta formation but it is likely that is also due to placental defects (and Sox2-Cre model would have answered it).

The study provides mechanism but the mechanism is somewhat less defined and blurry. The first part doesn't connect with the second part as the authors nicely acknowledge. Lineage specification is not affected but placental cell type function and architecture are affected. This is not due to early embryonic events, rather later events in the placenta. Not sure if de-repression of germ cell genes causes the defects in placenta or deregulation of placental genes. Likely both but the how part of it is unknown. There is

no elucidation on what type of transcription programs/genes are disrupted due to deregulation of placental genes and how that affects placental layers formation. Some comments on the roles of these deregulated genes in placental layer formation will be useful.

Thank you for the reviewer's comment; the Sox2-Cre *Dnmt3a/b* cDKO may indeed generate very interesting results and remains an ongoing effort in the lab. As suggested, we have added discussion about the value of this model for future studies to better explore the role of dysregulated gene expression and placental labyrinth formation (lines 308-309).

Reviewer #3 (Remarks to the Author):

Review: NCOMMS-22-22567-T

Title: Mechanisms and function of de novo DNA methylation in placental development

Comments to the authors:

The present study by Andrews et al investigated the impact of de novo DNMTs on the placenta trophoblast cell lineage during embryogenesis. This study makes important contributions to the field of placental epigenetics. In particular, the authors provide evidence to support that Dnmts 3A/B/L are essential for the establishment of the placental methylome, and that they exert their functions, in part, by targeting specific chromatin states. This highlights the interplay between different types of epigenetic modifications that are oftentimes studied in isolation. Additionally, this study demonstrates that a loss of Dnmt3B in particular is associated with lethality due to impaired placental labyrinth formation, with data suggesting this is due to transcriptional changes in genes typically expressed during gametogenesis, rather than issues with cell lineage specification or differentiation. This manuscript was well written, and easy to follow, and is appropriate for the audience of Nature Communications. The figures, extended figures, and supplemental tables were easy to interpret. Particular strengths of this study include the assessment of the interactions between DNA methylation and histone modifications, and the delineation of the important role of DNMTs, particularly DNMT3B in placental formation and function. This work is significant given the important public health concerns associated with placental dysfunction during pregnancy that the authors highlight. However, there are some suggested changes and clarifications that I would like the authors to address to strengthen their manuscript.

Major Comments:

1) While the methods section is very detailed, there is no statistical analysis section. Additionally, there are few statistical tests that were performed/reported in the presentation of the data. While the figures are clear and well-presented, the limited analysis makes it difficult to interpret the findings and draw accurate conclusions at times. Please address the following points about statistical analyses/presentation of data below:

- In the bean plots in Figure 2c the legend describes “enrichment” though no statistical test was performed to support this analysis and interpretation.

We have amended the axes in Fig. 2c, figure legend and results section (line 104) to “ChIP-seq signal” or “abundance of histone marks.” We have moved Extended data Fig. 2c, which quantitatively shows the enrichment of hypoDMRs at chromatin features to Fig. 2d and have added detailed statistical comparisons of these enrichments in Supplementary table 2, the methodology has been added to lines 520-523.

- Bean plots in Figure 2d analyzing methylation, no analysis was performed to help interpret/bolster the meaning of these findings.

This is a descriptive analysis (now Extended data Fig. 2a); to improve clarity, we have provided additional details on the context and meaning of these data (lines 104-108).

- There is an R value reported in Figure 3b, but there is no description of what type of regression was run to achieve this value. Also, is this an r or R²?

Fig. 3b shows a linear correlation and an R value is provided; we have added details of this analysis to the Methods statistical analysis section (line 544-545).

- Figure 3d, an r value is reported but no description of the regression that was run to derive the r value, and no p-value for the correlation is provided in figure legends. In the text (Line 148), a p-value is reported, but we don't know what type of statistic was run.

A Pearson correlation coefficient is provided and the p-value was ascertained using a linear correlation test statistic. These details have been added to the figure legend and methods (line 544-545).

- Extended data Figure 3a-c, please include type of regression run to generate R value. There are no error bars for extended data Figure 3g.

These figures show a Pearson Correlation coefficient R value; this has been added to the methods (line 544-545). Extended Date Fig. 3g is showing the proportion of the total number of probes that fall within each sub-class of ERV, hence there are no error bars in this analysis.

- Figure 4b, can you quantify the difference in IF intensity to provide a quantitative difference in MCT1/4 staining between WT and KO, and then run a statistical analysis on this difference? This would bolster the compelling finding that was presented in this Figure.

Thank you for this suggestion, we have now quantified the intensity of the immunofluorescence for MCT1/4 in *Dnmt3b* KO and WT and this has been added to Extended data Fig 6e. We have also included the quantitation for the *Dnmt3b* cKO and WT. These additional analyses have added significantly to our interpretation of the data.

- Extended data Figure 5f – is this a significant finding/pattern of higher expression in the KO or just an observation?

To enable statistical comparison of these data, we grouped cell types by labyrinth and junctional zone. Using Fisher's Exact test, we compared the proportion of up- and down-regulated genes in the *Dnmt3b* KO in contrast to WT, showing that methylation-sensitive germline genes were significantly up-regulated in both placental compartments. This analysis has been added to Fig. 4f and is described in the Methods statistical analysis section (line 549-550).

- Figure 5c, there are no error bars on the box plot even though these are frequencies calculated across litters.

Error bars have been added to Fig. 5c (now Fig. 5e).

- 2) The current study is focused on the role of de novo DNMTs 3A and B, and the requisite cofactor 3L. However, there is no mention of the maintenance enzyme DNMT1. This seems important given that cell division and proliferation are occurring as the placenta develops, and thus, any existing methylation marks would be maintained across cell divisions by DNMT1. Further, compensatory de novo

methylation by DNMT1 has been reported in certain situations, particularly repetitive elements, which were investigated in the present manuscript (see PMID 34140676 for reference). Given that DNMT1 is expressed in the placenta, please address why a DNMT1 KO was not included in this study and how this might limit or contribute to the interpretation of the findings presented in this manuscript.

Genomic imprinting is fundamental to placental formation and has been widely studied; thus, we excluded DNMT1 from this study because DNMT1 mutants lose all genomic imprinting (Hirasawa *et al.* 2008 *Genes & Dev*; 22:1607-16). We have sought to clarify this point further in our introduction (line 61).

As the reviewer highlights, it has recently been reported that DNMT1 can compensate for absence of the *de novo* DNMTs at a subset of repetitive elements (Haggerty *et al.* 2021 *Nat Struct Mol Biol*; 28:594-603). We have added publicly available datasets to our analysis to assess the potential contribution of DNMT to *de novo* DNA methylation in our datasets. DNA methylation patterns of the *Dnmt3a/b* DKO epiblast and ExE are remarkably similar pre-implantation inner cell mass (ICM) and trophectoderm (TE), with all samples clustering together by PCA (data added to Fig. 1b, lines 93-94). Hence, this analysis supports that without DNMT3A and DNMT3B there is no genome-wide 'wave of *de novo* methylation,' demonstrating that DNMT1 does not have a widespread capacity to *de novo* methylate; rather, as shown by Haggerty *et al.*, the *de novo* activity of DNMT1 appears to be largely restricted to a specific set of repetitive elements.

Minor Comments:

Introduction

Line 35: "DNA methylation is a repressive epigenetic...". While this is largely true, it is an over characterization/simplification of the role of DNA methylation in regulating transcription as there are times where more methylation is associated with increased gene expression. Please modify this language.

We have modified this statement (line 30).

Line 87: Can authors explain why they did not include a 3A/3L or 3B/3L double knockout in their studies?

Due to limited resources for mouse colony breeding, at the planning stages of the project we prioritised the *Dnmt3a/3b* DKO. We agree that 3A/3L or 3B/3L DKOs would be interesting for future studies.

Results

Lines 68-96: There is no mention of Figure 1d in the text.

This error has been corrected (line 676).

Line 85: Please explain why 20% methylation difference was chosen.

Using a power calculation, based on our sample size, we can detect >20% difference in DNA methylation between groups (p<0.05, 80% power, assuming +/-10% variation).

Lines 97-100 & 123-127: These types of summaries are more appropriate for the discussion than the results section.

Both statements have been amended to remove interpretation from the results section. These now provide only a succinct summary of the findings (lines 94-95, 122-124).

Line 148/Figure 3d: The x-axis is a little confusing for this figure. Consider plotting the WT and KO regressions separately, but overlapping on the same figure, to show the difference in the direction of the association based on Dnmt status.

We find it valuable to display *Dnmt* KOs on a single plot, as this demonstrates the continuum of de-repression that occurs upon loss of DNA methylation at these genes. We have rephrased the description of this plot in the results section (lines 143-145) and simplified Fig. 3d to improve clarity.

Lines 176-179: Sentence beginning “branching morphogenesis...” is a little unclear grammatically. Please edit.

The sentence has been edited to improve clarity (lines 180-182).

Discussion

Line 244: Specify this as a primary role of DNA methylation in the placenta.

The importance of DNA methylation in silencing germline genes has been reported across cell contexts, supporting that this is a widespread mechanism. We have cited these studies to support this statement (lines 251-252).

Can the authors expand upon why they observed more severe phenotypes with the *Dnmt3b* KO than the *Dnmt3a* knockout?

The difference in severity of the *Dnmt3b* and *Dnmt3a* KO placental phenotypes are consistent with those observed in the embryo: *Dnmt3b* KO embryos die at mid-gestation (E13.5) while *Dnmt3a* KO embryos survive until 3 weeks after birth (Okano *et al.* 1999 *Cell*; 99:247-57). Consistent with the DNA methylation losses we observe across KOs, DNMT3B is the primary *de novo* methyltransferase during embryogenesis.

Methods

See above about including statistical analysis section to the methods.

As suggested, we have added a statistical analysis section to the Methods (lines 543-553). We have also added further detail to the immunofluorescence (lines 405-406), bisulphite- (lines 509-511) , RNA-seq (lines 529-534) and UMAP analysis (lines 520-523) sections to provide better descriptions of our statistical methodology.

REVIEWERS' COMMENTS

Reviewer #1 (Remarks to the Author):

The authors have adequately responded to our comments, and the paper is much improved. We recommend for publication.

Reviewer #2 (Remarks to the Author):

The authors have addressed my comments. The manuscript is suitable for publication. There are some minor comments below that need to be fixed.

1. From the data provided it appears that 3b cKO mice do not survive. The data shows they make it to E18.5 (albeit smaller in weight) but the next time point is P10 where they don't make it. There are no pictures or weighs or viability of any postnatal animals. So it is better to accurately state that "3b cKO, in contrast to 3bKO, develop further to late gestation but are not viable (or viability is not studied here)"

2. Sentence on lines 308-309 on 3a/b cDKO is out of place. Please give it some context. The previous sentences don't discuss 3a (only talks 3b and Dnmt1) and suddenly this sentence about 3a/b cDKO appears.

3. The legend for Figure 5C (on page 27) is missing. Please add this.

Side note: Cyp19 is not a good Cre for trophoblast specific deletion as it is leaky in embryo proper later in embryogenesis. Several labs have struggled with this, so good that you didn't use it.

Reviewer #3 (Remarks to the Author):

Review of Revisions: NCOMMS-22-22567A

Title: Mechanisms and function of de novo DNA methylation in placental development reveals an essential role for DNMT3B

Comments to the authors:

The present revision by Andrews et al addressed the major and minor comments by all three reviewers in a satisfactory manner. The additional experiments and analyses performed, the updated methods and figures, and the expanded comments in the discussion strengthen the results and the interpretation of the findings. I would recommend this manuscript for publication. This work remains appropriate for the audience of Nature Communications and will add important information to the fields of epigenetics and placental biology.

RESPONSE TO REVIEWERS' COMMENTS

Reviewer #1 (Remarks to the Author):

The authors have adequately responded to our comments, and the paper is much improved. We recommend for publication.

Reviewer #2 (Remarks to the Author):

The authors have addressed my comments. The manuscript is suitable for publication. There are some minor comments below that need to be fixed.

1. From the data provided it appears that 3b cKO mice do not survive. The data shows they make it to E18.5 (albeit smaller in weight) but the next time point is P10 where they don't make it. There are no pictures or weighs or viability of any postnatal animals. So it is better to accurately state that "3b cKO, in contrast to 3bKO, develop further to late gestation but are not viable (or viability is not studied here)"

We have clarified the results presented in this section, and, as suggested, have highlighted that the phenotype underpinning the observed perinatal lethality before postnatal day 10 was not assessed in this study (lines 235-243).

2. Sentence on lines 308-309 on 3a/b cDKO is out of place. Please give it some context. The previous sentences don't discuss 3a (only talks 3b and Dnmt1) and suddenly this sentence about 3a/b cDKO appears.

Thank you, we have rephrased this sentence to improve clarity (lines 313-314).

3. The legend for Figure 5C (on page 27) is missing. Please add this.

We have added the legend for Figure 5C (lines 739-741).

Side note: Cyp19 is not a good Cre for trophoblast specific deletion as it is leaky in embryo proper later in embryogenesis. Several labs have struggled with this, so good that you didn't use it.

Reviewer #3 (Remarks to the Author):

Review of Revisions: NCOMMS-22-22567A

Title: Mechanisms and function of de novo DNA methylation in placental development reveals an essential role for DNMT3B

Comments to the authors:

The present revision by Andrews et al addressed the major and minor comments by all three reviewers in a satisfactory manner. The additional experiments and analyses performed, the updated methods and figures, and the expanded comments in the discussion strengthen the results and the interpretation of the findings. I would recommend this manuscript for publication. This work remains appropriate for the audience of Nature Communications and will add important information to the fields of epigenetics and placental biology.